# Redox activation of ATM enhances GSNOR translation to sustain mitophagy and tolerance to oxidative stress

Claudia Cirotti[1,2,†] , Salvatore Rizza[3,†] , Paola Giglio[1], Noemi Poerio[1], Maria Francesca Allega[3,‡],
Giuseppina Claps[4], Chiara Pecorari[3], Ji-Hoon Lee[5], Barbara Benassi[6], Daniela Barilà[1,2],
Caroline Robert[4,7,8], Jonathan S Stamler[9], Francesco Cecconi[1,10,11], Maurizio Fraziano[1],
Tanya T Paull[5] & Giuseppe Filomeni[1,3,12,*]

## Abstract

The denitrosylase *S*-nitrosoglutathione reductase (GSNOR) has
been suggested to sustain mitochondrial removal by autophagy
(mitophagy), functionally linking *S*-nitrosylation to cell senescence
and aging. In this study, we provide evidence that GSNOR is
induced at the translational level in response to hydrogen peroxide
and mitochondrial ROS. The use of selective pharmacological inhi-
bitors and siRNA demonstrates that GSNOR induction is an event
downstream of the redox-mediated activation of ATM, which in
turn phosphorylates and activates CHK2 and p53 as intermediate
players of this signaling cascade. The modulation of ATM/GSNOR
axis, or the expression of a redox-insensitive ATM mutant influ-
ences cell sensitivity to nitrosative and oxidative stress, impairs
mitophagy and affects cell survival. Remarkably, this interplay
modulates T-cell activation, supporting the conclusion that GSNOR
is a key molecular effector of the antioxidant function of ATM and
providing new clues to comprehend the pleiotropic effects of ATM
in the context of immune function.

**Keywords** ATM; GSNOR; mitophagy; ROS; T cell
**Subject Categories** Autophagy & Cell Death; Immunology; Molecular
Biology of Disease
2020 | Accepted 14 October 2020 | Published online 27 November 2020

## Introduction

Oxidative stress is a condition in which endogenously or exoge-
nously produced pro-oxidant species—e.g. reactive oxygen and
nitrogen species (ROS and RNS, respectively)—overwhelm the basal
antioxidant defense (Sies *et al*, 1985, 2017). The resulting alteration
of cell redox homeostasis usually induces a response aimed at inten-
sifying the expression, stabilization and activation of the following:
(i) primary antioxidants, required to intercept and scavenge ROS
and RNS, and (ii) repair machineries and cellular processes (e.g.,
DNA damage response, proteasome, autophagy) needed to fix any
injury that can compromise cell integrity (Halliwell & Gutteridge,
1995; Sies *et al*, 2017).

Dysfunctions in such response result in the accumulation of
damaged biomolecules and cellular structures, which is a hallmark
of cell senescence. In this scenario, mitochondria play a fundamen-
tal role as they are the main intracellular source of oxygen free
radicals and, on the other hand, one of the main targets of ROS
and RNS. This double nature implies that, once damaged upon
oxidative stress, mitochondria produce ROS at high rate, this fuel-
ing a vicious cycle that leads to cellular wasting and senescence
(Balaban *et al*, 2005).

We recently reported that the NADH-dependent denitrosylase *S*-
nitrosoglutathione reductase (GSNOR), by regulating the *S*-nitrosyla-
tion state of proteins involved in mitochondrial dynamics and mito-
phagy, sustains correct mitochondrial removal and delays cell

1  Department of Biology, Tor Vergata University, Rome, Italy
2  Laboratory of Cell Signaling, Istituto di Ricovero e Cura a Carattere Scientifico (IRCCS) Fondazione Santa Lucia, Rome, Italy
3  Redox Signaling and Oxidative Stress Group, Danish Cancer Society Research Center, Copenhagen, Denmark
4  INSERM, U981, Villejuif, France
5  Department of Molecular Biosciences, The University of Texas at Austin, Austin, TX,  USA
6  Division of Health Protection Technologies, ENEA-Casaccia, Rome, Italy
7  Université Paris Sud, Université Paris-Saclay, Kremlin-Bicêtre, France
8  Oncology Department, Gustave Roussy, Université Paris-Saclay, Villejuif, France
9  Institute for Transformative Molecular Medicine, Case Western Reserve University and Harrington Discovery Institute, University Hospitals Case Medical Center, Cleveland, OH,  USA
10  Cell Stress and Survival Unit, Danish Cancer Society Research Center, Copenhagen, Denmark
11  Department of Pediatric Hematology and Oncology, IRCCS Bambino Gesù Children's Hospital, Rome, Italy
12  Center for Healthy Aging, Copenhagen University, Copenhagen, Denmark
   *Corresponding author. Tel: +45 3525 7402; Emails: giufil@cancer.dk; filomeni@bio.uniroma2.it
   †These authors contributed equally to this work
   ‡Present address: Cancer Research UK Beatson Institute, Garscube Estate, Glasgow, UK

senescence (Rizza *et al*, 2018). Based on these lines of evidence, GSNOR activity extends beyond denitrosylation, having the potency to indirectly impinge on cellular redox state.

Similarly, ataxia telangiectasia (A-T) mutated (ATM), the DNA damage-responsive kinase acting as one of the early sensors of genome integrity, has been proposed to be also involved in the regulation of cellular antioxidant response (Barzilai *et al*, 2002; Ditch & Paull, 2012). *ATM*-null cell lines exhibit high rate of lipid peroxidation (Watters *et al*, 1999), reduced antioxidant response (Yi *et al*, 1990; Ward *et al*, 1994), and decreased GSH levels (Meredith & Dodson, 1987). Moreover, ATM-deficient mice show increased levels of ROS and signs of oxidative stress in some areas of the brain (Kamsler *et al*, 2001; Quick & Dugan, 2001). Taken together, these results suggest that oxidative stress conditions contribute to the A-T phenotype of defects in DNA repair. To strengthen this hypothesis, it has been reported that, besides recognizing and responding to DNA damage, ATM can also act as a direct $H_2O_2$ sensor. This additional ability is provided by the reactive Cys2991 that, once oxidized, resolves into an intermolecular disulfide bond bridging two ATM subunits (Guo *et al*, 2010) and triggers phosphorylation at Ser1981 through a specific signaling pathway (Lee *et al*, 2018).

Here, we have characterized a new $H_2O_2$-mediated ATM/GSNOR signaling axis whose primary function is to protect the cell against nitroxidative stress through fueling mitochondrial removal *via* mitophagy. Importantly, we demonstrate that this pathway plays a role in T-cell activation, supporting the biological significance of our findings and providing a molecular rationale for the conserved immunodeficient phenotype of ATM$^{-/-}$ and GSNOR-null organisms.

## Results

### GSNOR is post-transcriptionally induced by $H_2O_2$

By directly reducing GSNO, GSNOR indirectly regulates protein *S*-nitrosylation (Liu *et al*, 2001, 2004; Rizza & Filomeni, 2017), which has been recently demonstrated to keep mitophagy sustained, prevent oxidative damage, and delay cell senescence (Rizza *et al*, 2018). Based on this evidence, we hypothesized that, analogously to many antioxidant enzymes, GSNOR expression could be modulated in response to $H_2O_2$, whose intracellular concentration has frequently been found to increase in experimental models of aging (Sohal & Sohal, 1991; Balaban *et al*, 2005; Sohal & Orr, 2012). To verify our hypothesis, we treated HEK293 cells with non-toxic doses of $H_2O_2$ and evaluated GSNOR protein levels by Western blot analyses. Results shown in Fig 1A indicate that GSNOR is induced and its levels kept significantly sustained, up to 24 h of treatment with 100 μM $H_2O_2$.

We previously demonstrated that GSNOR is expressed concomitantly with Nrf2 activation in cellular models of amyotrophic lateral sclerosis and plays a role in the protection against NO donors-induced cell death (Rizza *et al*, 2015). By matching this evidence with data shown above, we checked whether Nrf2 was responsible for GSNOR upregulation. Western blot analyses confirmed that Nrf2 accumulated dose-dependently in HEK293 nuclear fractions after 8-h treatment with $H_2O_2$ (Fig 1B) following GSNOR trend. Consistently, we observed a significant increase in the mRNA of Nrf2 target genes, such as heme oxygenase 1

(HMOX-1) and glutamate:cysteine ligase (GCL), which was coherently prevented by treatment with trigonelline, an Nrf2 inhibitor (Fig 1C). Unexpectedly, RT–qPCR showed that GSNOR mRNA expression was not modulated by $H_2O_2$, at least up to 24-h treatment (Fig 1C and D). Consistently, GSNOR protein levels were not affected by Nrf2 inhibition (Fig 1E), indicating that GSNOR induction was unrelated to any (Nrf2-dependent or independent) transcriptional events induced by $H_2O_2$. To understand how GSNOR was upregulated, we blocked protein synthesis or, alternatively, proteasome-dependent degradation by pre-incubating the cells with cycloheximide (CHX) or with the proteasome inhibitor MG132, respectively. Western blot analyses indicated that GSNOR increase was abolished by CHX (Fig 1F) but not by MG132 (Fig 1G), supporting the conclusion that $H_2O_2$-induced GSNOR accumulation is not the result of enhanced protein stability but, reasonably, related to an increased rate of mRNA translation.

Of note, GSNOR mRNA has been previously reported to have two untranslated open reading frames (uORFs) upstream of the start codon where ribosomes stall (Kwon *et al*, 2001), this event being responsible for regulating GSNOR mRNA translation. The translation rate of uORF-containing genes is usually enhanced in response to different environmental stresses, e.g., oxidative stress (Spriggs *et al*, 2010). Therefore, to investigate whether GSNOR mRNA translation was affected by $H_2O_2$, we performed sucrose gradient fractionations of monosomes and polysomes from HEK293 lysates. The obtained polysome profile showed that $H_2O_2$ treatment caused an increase of 80S peak (Fig 1H), as previously reported (Li *et al*, 2010). However, RT–qPCR analyses performed after 8 and 24 h of treatment with $H_2O_2$ clearly showed a specific association of GSNOR mRNA with the heavy polysome fractions obtained by extracting mRNA from the last five fractions pooled together (Fig 1I). Cellular protein synthesis, evaluated by the method of puromycin—a structural analog of tyrosyl-tRNA—incorporation (Schmidt *et al*, 2009; Fig 1J) indicated that the overall translational profile did not significantly change in our experimental conditions, thus supporting the hypothesis that translation was selectively enhanced for GSNOR mRNA upon $H_2O_2$ exposure.

### GSNOR is target of ATM/CHK2 signaling pathway

Data from the literature demonstrate that ATM is a stress sensor involved in DNA double strand break (DSB) repair and in oxidative stress response (Guo *et al*, 2010; Tang *et al*, 2015). Therefore, in searching for the factor(s) regulating GSNOR expression, we focused on ATM. In particular, we checked for ATM auto-phosphorylation at Ser1981 (which is a marker of its activation), as well as the ATM-mediated phosphorylation of its downstream target, i.e., checkpoint kinase 2 (CHK2) at Thr68. As previously reported (Guo *et al*, 2010), Western blot analyses indicated that both these phosphorylations increased after $H_2O_2$ treatment (Fig 2A). To evaluate whether the ATM/CHK2 pathway was responsible for GSNOR modulation, we pharmacologically inhibited both the kinases and analyzed GSNOR levels by Western blot. Pre-incubation with KU55933 (KU) and AZD7762 (AZD), which was used to inhibit the kinase activity of ATM and CHK2, respectively, significantly prevented GSNOR increase induced by $H_2O_2$ (Fig 2B and C). Similar results were also obtained using siRNAs specifically targeting ATM or CHK2 (Fig 2D

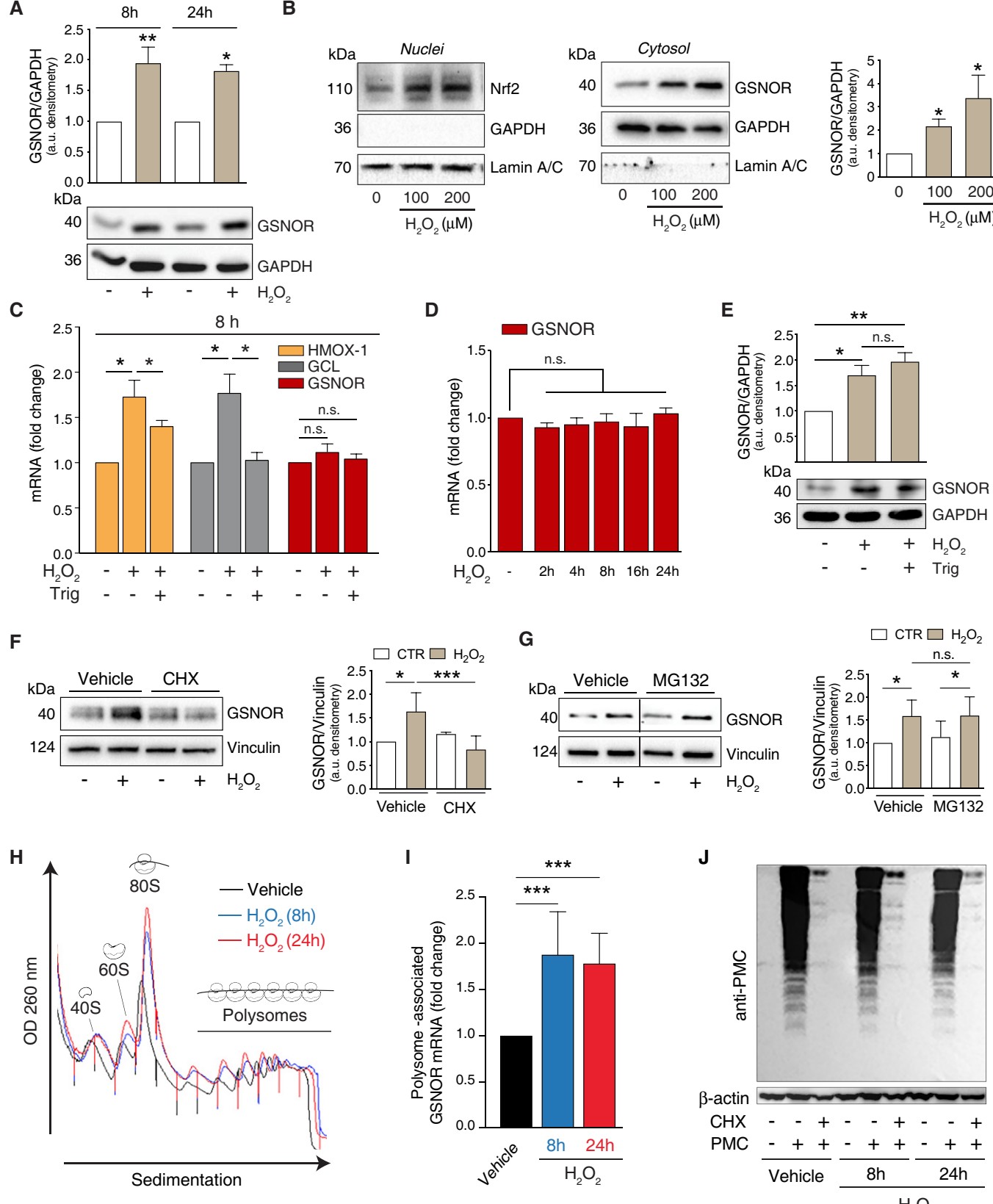

**Figure 1.**

**Figure 1.  GSNOR is translationally induced by hydrogen peroxide.**

A   HEK293 cells were treated for 8 and 24 h with 100 μM $H_2O_2$. GSNOR was evaluated by Western blot. Densitometry of each lane is normalized to GADPH, selected as loading control, and expressed as arbitrary units. Values shown represent the means ± SD of $n \geq 3$ independent experiments. *$P < 0.05$; **$P < 0.01$ calculated with regard to $H_2O_2$-untreated cells.

B   HEK293 cells were treated for 8 h with 100 or 200 μM $H_2O_2$. Nrf2 (left panel) and GSNOR (right panel) were evaluated by Western blot performed in nuclear and cytosol-enriched fractions. GAPDH and Lamin A/C were used as loading and purity controls of cytosol and nuclei, respectively. Densitometry of GSNOR immunoreactive bands is normalized to GADPH and expressed as arbitrary units. Values shown represent the means ± SD of $n = 3$ independent experiments. *$P < 0.05$.

C   HEK293 cells were treated for 8 h with 100 μM $H_2O_2$ in the presence or absence of the Nrf2 inhibitor trigonelline (*Trig*). Heme oxygenase 1 (HMOX-1), glutamate:cysteine ligase (GCL), and GSNOR expressions were evaluated RT–qPCR analyses Results shown are the means ± SEM of $n = 3$ experiments run in triplicate. *$P < 0.05$; *n.s.*, not significant.

D   RT–qPCR analyses of GSNOR mRNA after 2- to 24-h incubation with 100 μM $H_2O_2$. Results shown are the means ± SEM of $n = 3$ experiments run in triplicate, analyzed using ANOVA with Dunnett multiple comparisons test. *n.s.*, not significant.

E   Western blot analysis of GSNOR in HEK293 cells treated for 8 h with 100 μM $H_2O_2$ in the presence or absence of trigonelline (*Trig*). Densitometry of GSNOR immunoreactive bands is normalized to GADPH, selected as loading control, and expressed as arbitrary units. Values shown represent the means ± SD of $n = 3$ independent experiments. *$P < 0.05$; **$P < 0.01$.

F, G   HEK293 cells were treated for 8 h with 100 μM $H_2O_2$ in the presence or absence of in the presence or absence of cycloheximide (*CHX*, F), or the proteasome inhibitor MG132 (G). GSNOR was evaluated by Western blot. Vertical dotted lines represent a virtual division of the nitrocellulose filter, as immunoreactive bands reported in figure − although part of the same experiment/gel − were not contiguous. *Vehicle*: PBS. Densitometry of GSNOR immunoreactive bands is normalized to Vinculin, selected as loading control, and expressed as arbitrary units. Values shown represent the means ± SD of $n = 4$ independent experiments. *$P < 0.05$; ***$P < 0.001$; *n.s.*, not significant.

H   HEK293 cells were treated for 8 h (blue) or 24 h (red) with 100 μM $H_2O_2$. Polysome profile showing monosomes and polysomes was obtained from control (Vehicle) and $H_2O_2$ treated lysates (for either 8 or 24 h) by separation on 5–50% sucrose linear density gradient and collection using a gradient fractionation system.

I   RT–qPCR analyses of GSNOR mRNA in input and heavy polysome fraction pooled together in $H_2O_2$-treated samples (for either 8 or 24 h). H3A mRNA expression level was used as housekeeping gene control. Results represent the means ± SD of $n \geq 6$ independent experiments shown as % normalized by H3A. ***$P < 0.001$ *Vehicle*: PBS.

J   HEK293 cells treated for 8 h or 24 h with 100 μM $H_2O_2$ in the presence or absence of puromycin (tyrosyl-tRNA mimic used to label nascent proteins), or cycloheximide (*CHX*, used to block protein translation). Nascent polypeptides were assessed by Western blot performed with an anti-puromycin (*anti-PMC*) antibody. *Vehicle*: PBS. β-actin was used as loading control.

Source data are available online for this figure.

and E), strongly arguing for a role of ATM/CHK2 axis in GSNOR induction upon oxidative stress.

As GSNOR is the main enzyme responsible for the maintenance of *S*-nitrosylated proteins (PSNOs), we measured their levels as an assessment of GSNOR activity. In line with results shown above, $H_2O_2$-induced GSNOR upregulation correlated with a basal decrease of PSNOs levels, which accumulated upon the pharmacological inhibition of ATM (Fig 2F). Coherently, we did not detect any association of GSNOR mRNA with heavy polysome fractions (Fig 2G), eventually confirming that ATM acted as upstream modulator of GSNOR protein levels in response to $H_2O_2$ *via* enhancing its translation rate.

Finally, to exclude any possible role of ATM/CHK1 signaling pathway in GSNOR upregulation, we evaluated CHK1 phosphorylation at Ser317. This is an ATM-related (ATR) target residue, but can be also recognized and phosphorylated by ATM (Gatei *et al*, 2003). Western blot analyses demonstrated that there is only a slight increase in the levels of phospho-CHK1[Ser317] upon $H_2O_2$ treatment (Fig 2H). Moreover, GSNOR induction by $H_2O_2$ was not affected by CHK1 downregulation (Fig 2I), confirming that ATM/CHK1 axis was not significantly—or only marginally—involved in the phenomena observed.

## GSNOR is induced upon redox activation of ATM

From a mere chemical point of view, $H_2O_2$ is a mild oxidant which mostly induces DNA single strand breaks (SSB). As a consequence, it has been reported that most of $H_2O_2$-induced H2A.X phosphorylation at Ser139 (γH2A.X)—which is commonly accepted as a marker of DNA damage—is mediated by ATR (Katsube *et al*, 2014) and not

by ATM (Willis *et al*, 2013). Although the concentration of $H_2O_2$ used in our studies was low, immunofluorescence analyses revealed the presence γH2A.X (Fig 3A). This event was rapidly induced, but also efficiently turned off, as only low levels γH2A.X were detected after 8 h of treatment (Fig 3B), arguing for a successful activation of the DNA damage repair in HEK293 cells upon mild $H_2O_2$ challenge. We then evaluated the phospho-active levels of ATR and CHK1 at Ser345, which represents the signaling axis responsive to DNA SSB. Western blot analyses indicated that ATR/CHK1 pathway (as well as ATM/CHK2) was activated soon after 1 h (Fig EV1A). However, they were no more detected neither after 8 nor after 24 h of treatment with $H_2O_2$ (Fig 3C), indicating that, at variance with ATR/CHK1 (whose induction was rapid but transient), the phospho-activation of ATM/CHK2 signaling axis was kept sustained even at longer time points, when H2A.X was no longer phosphorylated and DNA damage efficiently repaired. These results gave further strength to the idea that, in our experimental conditions, ATM persistent phosphorylation underpinned functions which were associated with GSNOR induction but, probably, not related to DNA damage repair. In support to this hypothesis, Western blot analyses of GSNOR indicated that it did not increase at very early time points, but started to be upregulated only after 4 h of $H_2O_2$ treatment (Fig EV1B), when H2A.X phosphorylation reached the highest level and began to decline (Fig EV1C).

To finally exclude any involvement of the DNA damage response on GSNOR induction, we treated HEK293 cells with neocarzinostatin (NCS), a well-known DNA damaging radiomimetic, and evaluated the basal and phospho-active levels of ATM/CHK2 and ATR/CHK1 signaling axes. Western blot analyses showed that all these molecular factors were phosphorylated, along with the phosphorylation of

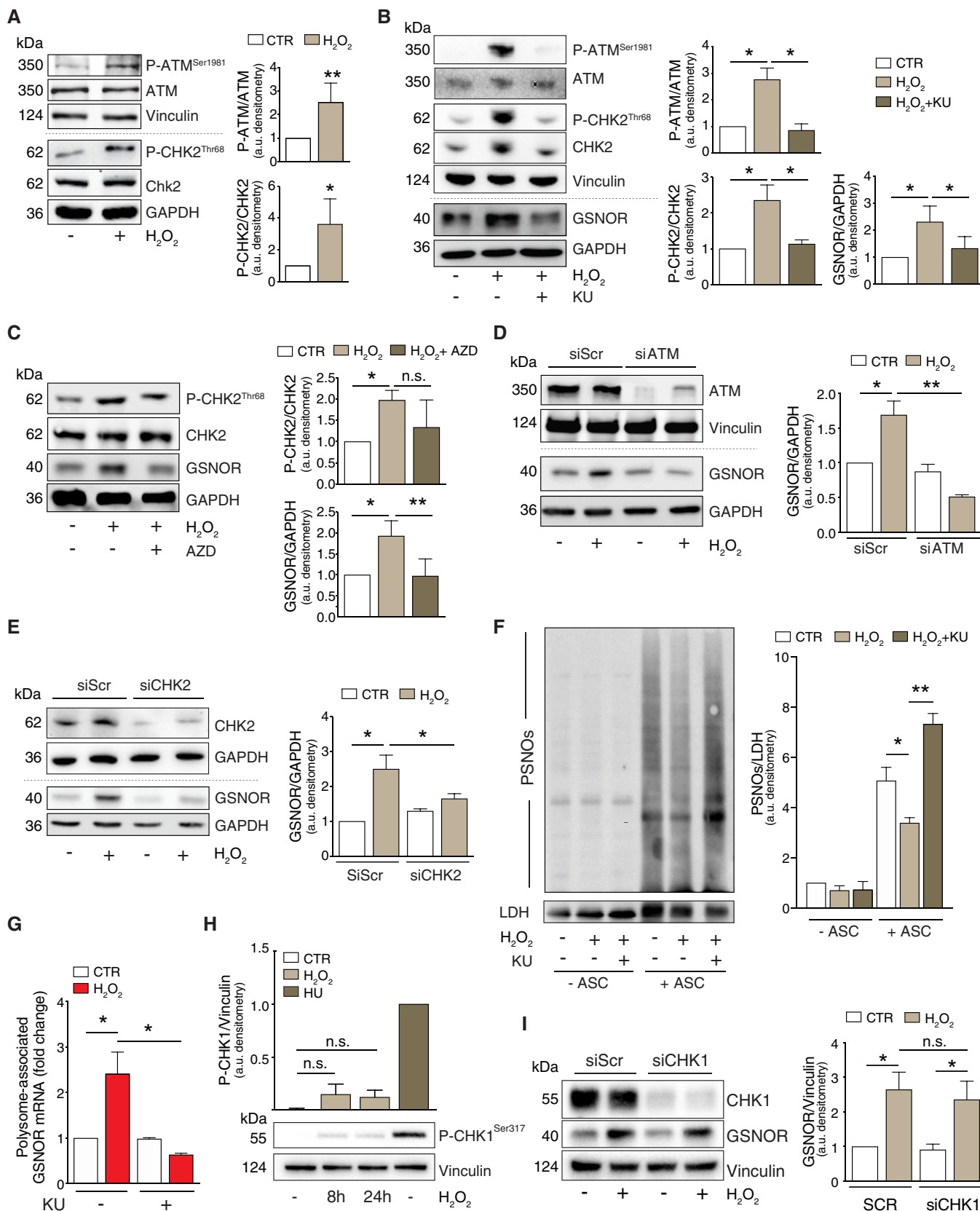

**Figure 2.**

◀ **Figure 2. GSNOR induction by hydrogen peroxide occurs *via* the phospho-activation of ATM/CHK2 signaling axis.**

A   HEK293 cells treated for 24 h with 100 μM $H_2O_2$. Basal and phospho-active forms of ATM and CHK2 were assessed by Western blot. Phospho:basal level ratios of ATM and CHK2 immunoreactive bands are normalized to Vinculin, selected as loading control, and expressed as arbitrary units. Values shown represent the means ± SD of $n \geq 4$ independent experiments. *$P < 0.05$; **$P < 0.01$.

B   HEK293 cells were treated for 24 h with 100 μM $H_2O_2$ in the presence or absence of the ATM inhibitor KU55933 (*KU*). Basal and phospho-active forms of ATM, CHK2, and GSNOR were assessed by Western blot. Vinculin and GAPDH were used as loading controls. Phospho:basal level ratios of ATM and CHK2 (normalized to Vinculin) along with densitometry of GSNOR immunoreactive bands (normalized to GAPDH) are expressed as arbitrary units. Values shown represent the means ± SD of $n = 3$ independent experiments. *$P < 0.05$.

C   HEK293 cells were treated for 24 h with 100 μM $H_2O_2$ in the presence or absence of the CHK2 inhibitor AZD7762 (*AZD*). Basal and phospho-active forms of CHK2, together with GSNOR, were assesses by Western blot. Phospho:basal level ratios of CHK2 and densitometry of GSNOR immunoreactive bands are normalized to GAPDH, selected as loading control, and expressed as arbitrary units. Values shown represent the means ± SD of $n = 3$ independent experiments. *$P < 0.05$; **$P < 0.01$; *n.s.*, not significant.

D   HEK293 cells were transfected with a pool of siRNA against ATM (siATM) or with control siRNA (siScr) for 18 h and treated for additional 24 h with 100 μM $H_2O_2$. Basal and phospho-active forms of ATM, together with GSNOR were assessed by Western blot. Vinculin and GAPDH were used as loading controls. Densitometry of GSNOR immunoreactive bands is normalized to GAPDH and expressed as arbitrary units. Values shown represent the means ± SD of $n = 3$ independent experiments. *$P < 0.05$; **$P < 0.01$.

E   HEK293 cells were transfected with a pool of siRNA against CHK2 (siCHK2) or with control siRNA (siScr) for 18 h and treated for additional 24 h with 100 μM $H_2O_2$. Basal and phospho-active forms of CHK2, together with GSNOR, were assessed by Western blot. GAPDH was used as loading control. Densitometry of GSNOR immunoreactive bands is normalized to GAPDH and expressed as arbitrary units. Values shown represent the means ± SD of $n = 3$ independent experiments. *$P < 0.05$.

F   HEK293 cells were treated for 24 h with 100 μM $H_2O_2$ in the presence or absence of the ATM inhibitor KU55933 (*KU*). Lysates were subjected to biotin-switch assay, and *S*-nitrosylated proteins (PSNOs) revealed by incubation with horseradish peroxidase (HRP)-conjugated streptavidin. Results obtained in the absence of ascorbate (- ASC) are shown as negative controls. Densitometry of each lane intensity is normalized to LDH, selected as loading control, and expressed as arbitrary units. Values shown represent the means ± SD of $n = 3$ independent experiments. *$P < 0.05$; **$P < 0.01$.

G   RT–qPCR analyses of GSNOR mRNA in input and heavy polysome fraction pooled together in $H_2O_2$-treated samples in the absence or in the presence of the ATM inhibitor KU55933 (*KU*). H3A mRNA expression level was used as housekeeping gene control. Results are the means ± SEM of $n = 3$ independent experiments run in duplicate and shown as %, normalized by H3A. *$P < 0.05$.

H   HEK293 cells were treated for 8 or 24 h with 100 μM $H_2O_2$ or, alternatively, for 2 h with 2 mM hydroxyurea (*HU*), selected as positive control of CHK1 phosphorylation at Ser317. Phospho-CHK1$^{Ser317}$ was assessed by Western blot; densitometry normalized to Vinculin, selected as loading control, and expressed as arbitrary units (with HU arbitrarily set to 1). Values shown represent the means ± SD of $n = 3$ independent experiments. *n.s.*, not significant.

I   HEK293 cells were transfected with a pool of siRNA against CHK1 (siCHK1) or with control siRNA (siScr) for 18 h and treated for additional 24 h with 100 μM $H_2O_2$. CHK1 and GSNOR were assessed by Western blot analysis of. Densitometry of GSNOR immunoreactive bands is normalized to Vinculin, selected as loading control, and expressed as arbitrary units. Values shown represent the means ± SD of $n = 3$ independent experiments. *$P < 0.05$.

Source data are available online for this figure.

p53 at Ser15 but, importantly, GSNOR did not accumulate (Fig 3D). In line with our hypothesis that GSNOR is induced by oxidative conditions, $H_2O_2$ intracellular levels did not increase upon NCS treatment (Fig 3E).

The DNA damage- and the redox-dependent activation mechanisms of ATM have been recently demonstrated to proceed through distinct signaling pathways (Lee *et al*, 2018). To investigate whether, in our conditions, ATM activation was a redox-dependent event, we overexpressed, in HEK293 cells, the *wild-type* (ATM$^{WT}$) or the redox mutant of ATM (ATM$^{CL}$), in which the Cys2991—which acts as redox center of the protein—was replaced by Leu (C2991L). Western blot analyses indicated that cells expressing the ATM$^{CL}$ mutant showed a weaker phosphorylation of ATM in response to 8-h treatment with $H_2O_2$ with respect to that observed in ATM$^{WT}$ expressing cells (Fig 3F). This result well correlated with GSNOR protein levels that—though higher than non-transfected cells—were not significantly modulated in ATM$^{CL}$ cells (Fig 3F). These results further argued for a redox-dependent, rather than DNA damage-induced activation of ATM being responsible for GSNOR induction in our experimental conditions.

### GSNOR induction is dependent on p53

Many lines of evidence indicate that p53 represents a multifunctional redox-sensing transcription factor, which can also play a role in post-transcriptional regulation of gene expression (Ewen &

Miller, 1996). In particular, it has been reported that ATM/p53 axis regulates the stability of a set of mRNAs upon ionizing radiation (Venkata Narayanan *et al*, 2017), which is also a condition associated with oxidative stress. Therefore, we first checked whether p53 was involved in GSNOR modulation. Western blot analyses indicated that, along with ATM and CHK2, p53 was phosphorylated at Ser15 upon $H_2O_2$ treatment (Fig 4A). To better investigate this aspect, we alternatively treated the cells with pifithrin-α (*Pft*)—one of the most used inhibitors of p53—or transfected them by siRNA against p53 (sip53) and demonstrated that GSNOR was no longer modulated by $H_2O_2$ (Fig 4A and B), although ATM and CHK2 were, in both cases, still active. Western blot analyses of p53 phosphorylation at Ser15, carried out upon ATM or CHK2 inhibition, confirmed that p53 activation was dependent on ATM and CHK2 (Appendix Fig S1A). Therefore, to go into much detail into the role of p53 in GSNOR modulation, we took advantage of a p53-null cellular model, i.e., the human colon carcinoma HCT116 cell line that we reconstituted with the *wild-type* form of p53 (p53$^{wt}$). Western blot analyses showed no GSNOR modulation in HCT116 cells upon treatment with $H_2O_2$ (Fig 4C), even under conditions in which CHK2 and ATM were activated. On the contrary, p53$^{wt}$ cells exhibited a greater amount of GSNOR, even without any prooxidant stimuli. This correlated with a significant GSNOR accumulation upon $H_2O_2$ treatment (Fig 4C), confirming the pivotal role for p53 in GSNOR modulation.

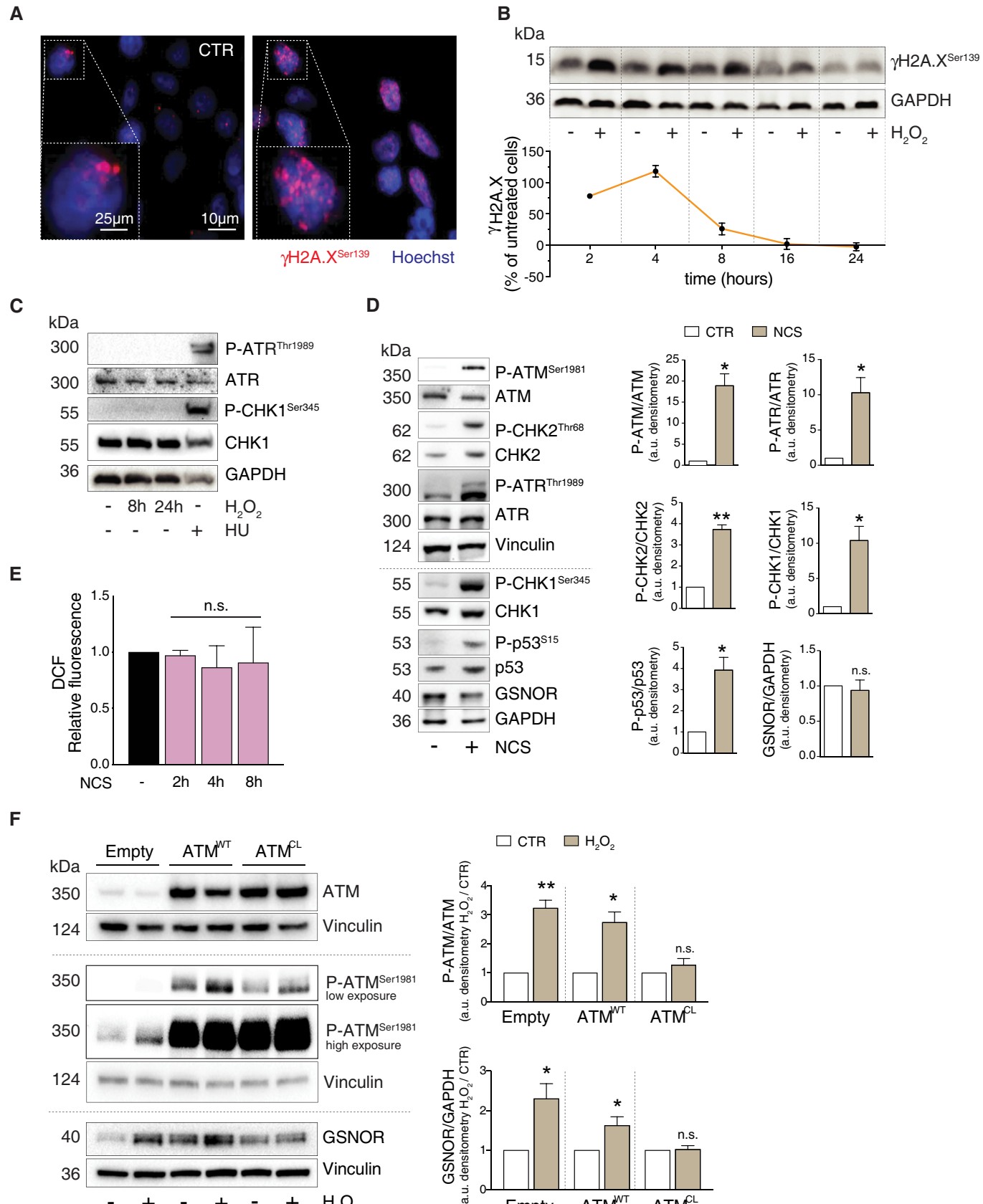

**Figure 3.**

Figure 3.   GSNOR induction by hydrogen peroxide is dependent on the redox activation of ATM and kept sustained over time.

A   HEK293 cells treated for 4 h with 100 μM $H_2O_2$. Phospho-histone H2A.X (γH2A.X) was assessed by immunofluorescence analysis. Nuclei (blue) were stained with Hoechst 33342. Scale bar: 10 μM. 4X digital magnification is shown at the bottom left of each image. CTR: $H_2O_2$-untreated cells.
B   Additionally, γH2A.X was also assessed by Western blot at the indicated time points. GAPDH was used as loading control (top panel). Densitometry of each lane intensity is normalized to GAPDH, selected as loading control, and expressed as % of γH2A.X in $H_2O_2$-treated versus untreated cells. Values represent the means ± SD of two independent experiments (bottom panel).
C   HEK293 cells were treated for 8 or 24 h with 100 μM $H_2O_2$ or, alternatively, for 2 h with 2 mM hydroxyurea (HU), selected as positive control of ATR/CHK1 axis activation. Basal and phospho-active forms of ATR and CHK1 (at Ser345) were assessed by Western blot. GAPDH was used as loading control.
D   HEK293 cells were treated for 8 h with 20 μM neocarzinostatin (NCS). (left) Western blot analysis of basal and phospho-active forms of ATM, CHK2, ATR, CHK1, p53, and GSNOR in HEK293 cells treated for 8 h with 20 μM neocarzinostatin (NCS). Vinculin and GAPDH were used as loading controls. (right) Phospho:basal level ratios of ATM, CHK2, ATR, and CHK1, along with densitometry of GSNOR immunoreactive bands are expressed as arbitrary units. Values shown represent the means ± SD of $n = 3$ independent experiments. *$P < 0.05$; **$P < 0.01$; n.s., not significant.
E   HEK293 cells were treated for 2, 4 and 8 h with 20 μM neocarzinostatin (NCS). After treatment, cells were incubated with 2′,7′-$H_2$DCF-DA to cytofluorometrically assess the intracellular production of $H_2O_2$. Values are shown as units of DCF fluorescence relative to untreated cells. Values are shown as units of fluorescence relative to NCS-untreated cells (arbitrarily set as 1) and represent the means ± SD of $n = 3$ independent experiments. n.s., not significant.
F   HEK293 cells were transfected with plasmids coding for the wild-type (WT), C2991L redox mutant (CL) of ATM, or with an empty vector (Empty; used as negative control) for 40 h and treated for additional 8 h with 100 μM $H_2O_2$. Basal and phospho-active forms of ATM and GSNOR were assessed by Western blot analysis. Vinculin and GAPDH were used as loading controls. Two different exposures (low and high) were selected to highlight differences in P-ATM levels in different experimental settings. Phospho:basal level ratios of ATM, along with densitometry of GSNOR immunoreactive bands are expressed as arbitrary units relative untreated cells. Values shown represent the means ± SD of $n = 3$ independent experiments. *$P < 0.05$; **$P < 0.01$; n.s., not significant.

Source data are available online for this figure.

Next, we selected the osteogenic carcinoma SAOS (Fig EV2A), which is an established p53-null model of cervix carcinoma and compared them with HeLa and osteosarcoma U2OS (Fig EV2B) as representative of p53 wild-type cell lines. Cells were treated with 100 μM $H_2O_2$ which, in the case of SAOS, were also increased up to 200 μM. Notwithstanding ATM and CHK2 phosphorylation was an event shared among all cell lines, GSNOR was induced in those expressing p53 (HeLa and U2OS) and not modulated in p53-null cells (SAOS), indicating that ATM/CHK2/p53/GSNOR signaling pathway is activated in multiple cellular models. To further strengthen this hypothesis, we performed similar experiments in immortalized human fibroblasts BJ-hTERT and observed similar trends (i.e., P-ATM and GSNOR increase) upon treatment with different concentrations of $H_2O_2$ (Appendix Fig S1B).

## GSNOR modulates mitophagy induced by $H_2O_2$ via ATM

It is well documented that low doses of $H_2O_2$ target mitochondria and result in mitochondrial impairment and mitophagy (Frank et al, 2012). To verify that mitochondria were damaged by $H_2O_2$, even in our cell system, we assessed the intracellular production of $H_2O_2$ and mitochondrial superoxide. Cytofluorometric analyses confirmed that intracellular concentration of ROS significantly increased when HEK293 cells were treated with 100 μM $H_2O_2$ (Fig 5A and B). To investigate whether this was related to mitophagy induction, we performed Western blot analyses of different mitochondrial complex subunits, along with RT–qPCR quantitation of D-loop, which gives an esteem of the mitochondrial DNA (mtDNA). Results obtained confirmed that $H_2O_2$ treatment caused a decrease of mitochondrial proteins (Fig 5C and D) and DNA (Fig 5E), which argued for a selective removal of mitochondria by mitophagy. Strikingly, both ATM and GSNOR silencing, elicited by siRNA, were able to prevent this process (Fig 5C–E). Actually, ATM knockdown induced per se a reduction of mitochondrial complexes, which is in line with a general decrease of mitochondrial mass, already reported in ATM-deficient human cell lines (Eaton et al, 2007). This gave strength to the emerging concept that—besides the effects on GSNOR—ATM is deeply (and more intimately) involved in mitochondrial homeostasis (Lee & Paull, 2020).

It has been reported that redox-insensitive $ATM^{CL}$ mutant does not compromise the efficiency of DNA damage repair but leads to a general dysfunction of mitochondria which are not selectively removed by mitophagy (Lee et al, 2018; Zhang et al, 2018). Similar phenotype was observed in $Gsnor^{-/-}$ mice (Rizza et al, 2018), in which it has been recently demonstrated that mitophagy is impaired, due to excessive S-nitrosylation (Rizza et al, 2018; Rizza & Filomeni, 2018). Based on these lines of evidence, we wanted to better investigate whether GSNOR acted as a downstream effector of ATM-driven mitophagy upon oxidative stress. To this end, we took advantage of a well-established cell system we have recently generated (Lee et al, 2018), i.e., U2OS cells stably expressing doxycycline-inducible alleles coding for the wild type ($ATM^{WT}$), the DNA damage unresponsive ($ATM^{2RA}$), or the redox-insensitive ($ATM^{CL}$) forms of ATM. To minimize the effects of endogenous ATM, before doxycycline treatment, the endogenous ATM was knocked down by shRNA (shATM cells), and GSNOR was concomitantly overexpressed to assess its direct role in mitophagy (Appendix Fig S2). Cells were then treated with $H_2O_2$ for 8 h, and mitophagy evaluated by both D-loop relative quantitation and confocal microscopy assessment of LC3 and TOM20—selected as autophagosome and mitochondrial markers, respectively. The latter analysis was performed upon incubation with chloroquine to inhibit autophagosome/lysosome fusion (Mauthe et al, 2018). This approach was used to emphasize co-localization between mitochondria and autophagosomes, which is a measure of a working mitophagy. Upon $H_2O_2$ treatment, mtDNA significantly decreased in $ATM^{WT}$ and $ATM^{2RA}$, but not in $ATM^{CL}$ cells, unless GSNOR was concomitantly expressed (Fig 5F). This result clearly indicated that mitophagy was sustained only if cells expressed GSNOR or a redox-sensitive form of ATM, confirming the role of the ATM/GSNOR axis as mitophagy tuner. Similarly, confocal microscopy images showed that mitochondrial co-localization with autophagosomes was induced in $ATM^{WT}$, but not in $ATM^{CL}$ cells challenged with $H_2O_2$, this condition being rescued when $ATM^{CL}$ cells overexpressed GSNOR (Fig 5G and H). Consistently, cells expressing $ATM^{2RA}$, which is normally activated by $H_2O_2$, but no longer able to respond to DNA damage (Lee et al, 2018), showed no difference with $ATM^{WT}$ cells in terms of mitophagy (Fig 5G and H).

## GSNOR modulates mitophagy induced by mitochondrial oxidative stress via ATM

To verify whether ATM/GSNOR-sustained mitophagy was a mechanism of general application or selectively induced by oxidative stress, we challenged mitochondria with different stimuli. First, we subjected HEK293 cells to hypoxia conditions (i.e., $pO_2 = 1\%$) for 4

and 8 h. D-loop quantitation and Western blot analyses of mitochondrial proteins indicated that mitophagy was efficiently induced by hypoxia (Fig EV3A and B). However, in the absence of any re-oxygenation, this condition was not associated with ROS production —actually, it was accompanied by a significant decrease of intracellular ROS levels (Fig EV3C)—neither with ATM/GSNOR axis activation (Fig EV3D). Similarly, 4- and 8-h treatment with a combination

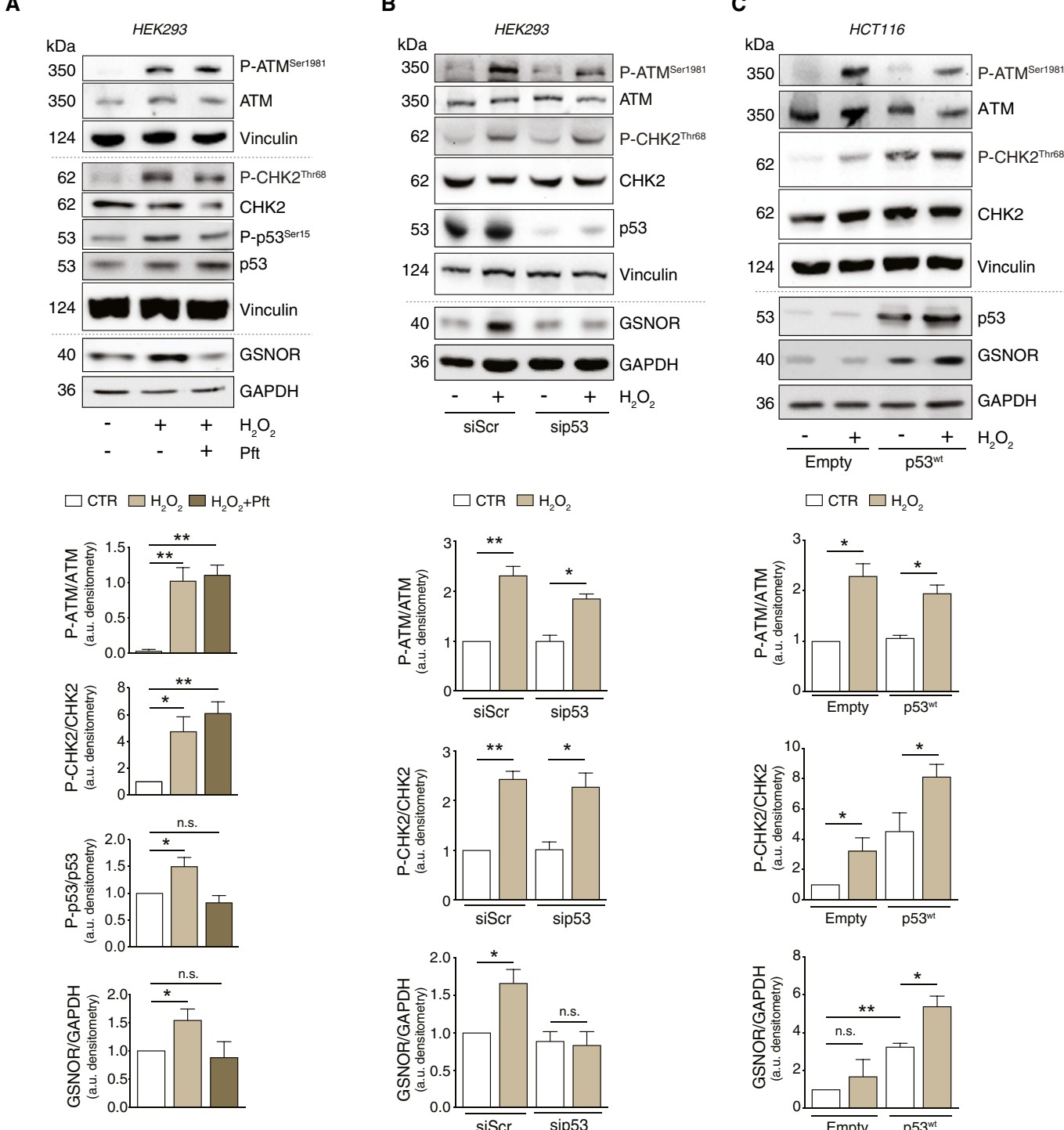

**Figure 4.**

**Figure 4. GSNOR induction by hydrogen peroxide involves p53 as downstream effector of ATM/CHK2 phosphorylation axis.**

A HEK293 cells were treated for 24 h with 100 μM $H_2O_2$ in the presence or absence of 20 μM pifithrin-α (*Pft*). Basal and phospho-active forms of ATM, CHK2, p53, and GSNOR were assessed by Western blot. Phospho:basal level ratios of ATM, CHK2, and p53 (normalized to Vinculin), along with densitometry of GSNOR immunoreactive bands (normalized to GAPDH) are expressed as arbitrary units. Values shown represent the means ± SD of *n* = 3 independent experiments. *$P < 0.05$; **$P < 0.01$; *n.s.*, not significant.

B HEK293 cells were transfected with a pool of siRNA against p53 (sip53) or with control siRNA (siScr) for 24 h and treated for additional 24 h with 100 μM $H_2O_2$. Basal and phospho-active forms of ATM and CHK2, as well as p53 and GSNOR, were assessed by Western blot. Phospho:basal level ratios of ATM and CHK2 (normalized to Vinculin), along with densitometry of GSNOR immunoreactive bands (normalized to GAPDH) are expressed as arbitrary units. Values shown represent the means ± SD of *n* = 3 independent experiments. *$P < 0.05$; **$P < 0.01$; *n.s.*, not significant.

C HCT116 cells expressing the *wild-type* form of p53 (p53$^{wt}$), or an empty vector (*Empty*, selected as negative control), were treated for 24 h with 100 μM $H_2O_2$. Basal and phospho-active forms of ATM and CHK2, p53, and GSNOR were assessed by Western blot. Vinculin and GAPDH were used as loading controls. Phospho:basal level ratios of ATM and CHK2, along with densitometry of p53 and GSNOR immunoreactive bands are expressed as arbitrary units. Values shown represent the means ± SD of *n* = 3 independent experiments. *$P < 0.05$; **$P < 0.01$; *n.s.*, not significant.

Source data are available online for this figure.

of antimycin and oligomycin, used at 1 μM to inhibit mitochondrial OXPHOS, efficiently triggered mitophagy (Fig EV3E and F). However, it resulted in ROS decrease and no changes in phospho-ATM and GSNOR levels (Fig EV3G and H). Next, we challenged mitochondrial homeostasis by carbonyl cyanide m-chlorophenylhy-drazone (CCCP)—an uncoupling molecule used to induce mito-phagy *in vitro*. As already reported elsewhere (Lopez-Fabuel *et al*, 2016), we observed that treatment with 10 μM CCCP resulted in ROS production (i.e., $H_2O_2$ and superoxide) even in HEK293 cells (Fig EV3I and J). This phenomenon was associated with a signifi-cant decrease in mtDNA content and prevented if ATM and GSNOR were previously knocked down through siRNA (Fig EV3K). Alto-gether, these data supported the hypothesis that ATM/GSNOR signaling axis was required to sustain mitophagy exclusively in response to oxidative stress.

Based on these results, we moved to U2OS cell and confirmed that CCCP elicited $H_2O_2$ and mitochondrial superoxide production (Fig 6A and B). This phenomenon was associated with the phos-pho-activation of ATM and the induction of GSNOR (Fig 6C), which was consistent with results so far obtained in cells treated with $H_2O_2$. Next, to verify that this was associated with to mitophagy, we induced the expression of ATM$^{WT}$ or ATM$^{CL}$ by doxycycline treat-ment in cells previously depleted of endogenous ATM (shATM cells), as above described. Here, we evaluated mtDNA content and followed the degradation of different subunits of mitochondrial respiratory proteins upon 8-h treatment with 10 μM CCCP. As previ-ously observed for $H_2O_2$, CCCP induced a significant decrease of mtDNA in ATM$^{WT}$ and ATM$^{2RA}$, but not in ATM$^{CL}$ cells, indicating that mitophagy was impaired when ATM was unable to directly respond to ROS (Fig 6D). Remarkably, GSNOR overexpression significantly reverted this phenotype and contributed to make ATM$^{CL}$ cells proficient in removing mitochondria (Fig 6D). These results were confirmed by confocal microscopy and Western blot analyses. Mitochondria/autophagosome co-localization (Fig 6E and F) and the levels of mitochondrial proteins (Fig 6G and H) decreased in ATM$^{WT}$ cells, but were not (or only slightly) affected in ATM$^{CL}$ variants, unless they expressed GSNOR. Conversely, cells carrying ATM$^{2RA}$ mutant were similar to ATM$^{WT}$ counterpart (Fig EV4A and B), suggesting that redox activation of ATM is specifically required for GSNOR-mediated mitochondrial removal.

## GSNOR protects from nitroxidative stress

Results so far obtained let to assume that GSNOR affected cellular redox homeostasis by: (i) regulating protein denitrosylation, and (ii)

removing mitochondria by mitophagy, suggesting it can counteract deleterious effects of NO and ROS. Along this line of reasoning, we wanted to investigate the consequences of GSNOR modulation in cases of combined insult of $H_2O_2$ and NO. To this end, we used HEK293 cells transiently overexpressing GSNOR (GSNOR$^{wt}$ cells) and performed combined treatments with $H_2O_2$ and the NO donor DPTA-NONOate (DPTA; see scheme in Fig 7A), both provided at doses that were not toxic *per se* (Appendix Fig S3). After treatment, we evaluated dead cells by direct cell counting using Trypan blue exclusion assays. As shown in Fig 7B, most of GSNOR$^{wt}$ cells and control counterparts carrying the empty vector (*Empty* cells) were viable, with low extent of dead cell. This argued for appropriate mechanisms of defense being activated. However, when we performed the same treatment in the presence of the ATM inhibitor, used to prevent GSNOR induction, *Empty* cells showed a consider-able increase in cell death (Fig 7B and C) occurring contextually with caspase3 and PARP1 disappearance (Fig 7D). Conversely, GSNOR$^{wt}$ cells did not show any significant change in the amount of dead cell and any modulations of apoptotic markers (Fig 7B–D).

We previously demonstrated that GSNOR sustains mitophagy through the active denitrosylation of Parkin (Rizza & Filomeni, 2018; Rizza *et al*, 2018, 2020). To investigate whether GSNOR-mediated protection toward $H_2O_2$ and NO toxicity was related to its effects on Parkin, we knocked down Parkin (Appendix Fig S4) in both GSNOR$^{wt}$ and control (*Empty*) cells and evaluated cell viability by Alamar blue assays upon treatment with $H_2O_2$ and DPTA. Fluorometric analyses indicated that Parkin downregulation significantly prevented GSNOR-mediated protection (Fig 7E), strengthening the hypothesis that Parkin-mediated mitophagy is the process through which GSNOR guarantees cell viability in nitroxidative stress conditions.

To confirm these results, we utilized mouse fibroblasts (MAFs) obtained from Gsnor *wild-type* (WT) and null (KO) mice. Western blot analyses substantiate that Gsnor was induced in WT MAFs treated with $H_2O_2$ (Fig 7F). In agreement with our hypothesis, Gsnor-null fibroblasts appeared to be highly susceptible to the combined treatment with DPTA and $H_2O_2$ (Fig 7G and H), reinforc-ing the idea of general protective role of GSNOR in cell response to nitroxidative stress.

## The ATM/GSNOR axis is involved in T-cell activation

Coexistence of $H_2O_2$ and NO fluxes is a condition that many cell types (e.g.: leukocytes or neurons) have to face and adapt to during their life. In particular, B and T lymphocytes have been reported to

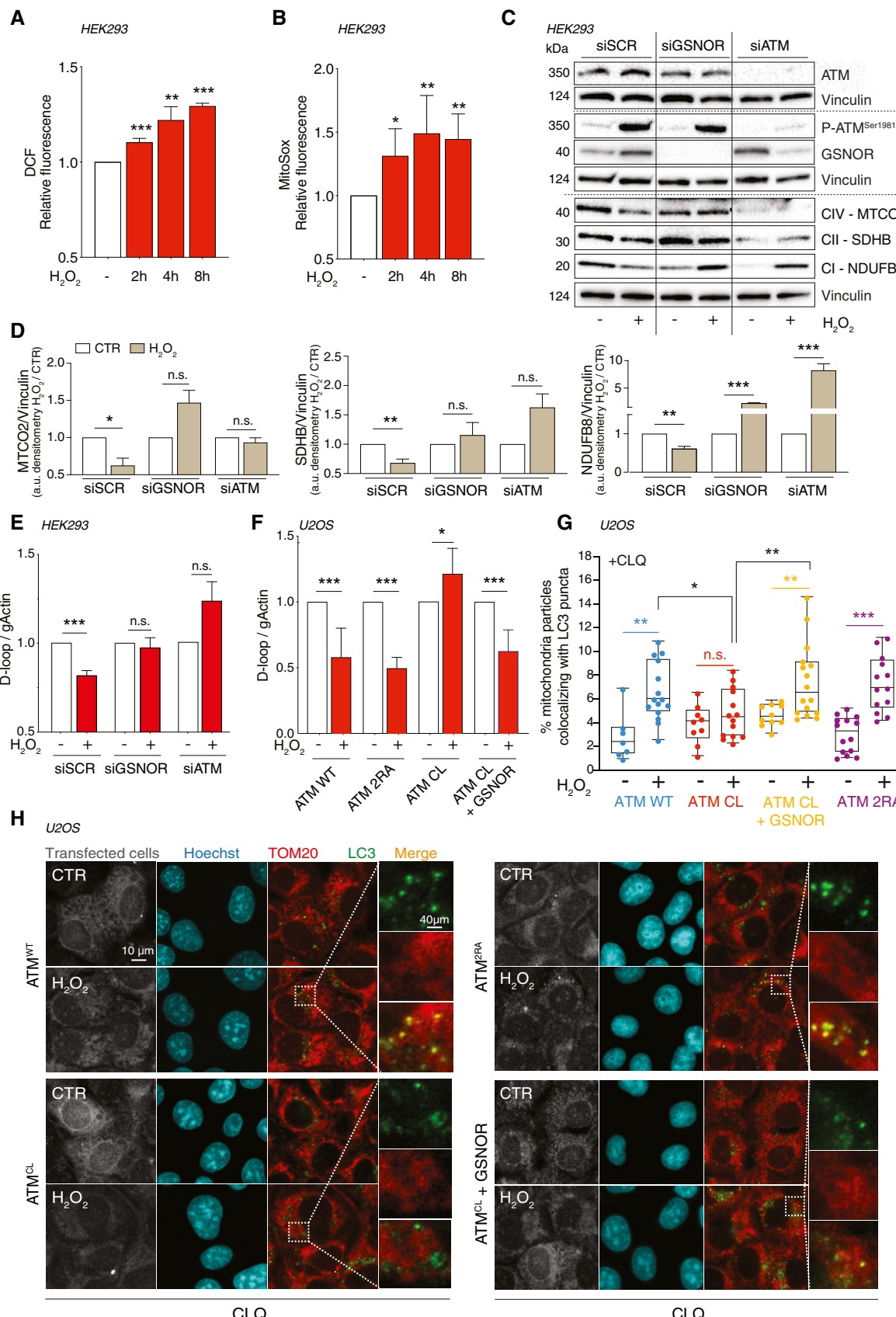

**Figure 5.**

**Figure 5. ATM/GSNOR axis drives mitophagy upon H$_2$O$_2$ treatment.**

A, B    HEK293 cells treated for 2, 4, and 8 h with 100 μM H$_2$O$_2$. After treatment, cells were incubated with 5 μM 2',7'-H$_2$DCF-DA (A) or MitoSox (B) to evaluate the production of H$_2$O$_2$ or mitochondrial superoxide, respectively. Values are shown as units of DCF or MitoSox fluorescence relative to untreated cells (arbitrarily set as 1) and represent the means $\pm$ SD of $n \geq 3$ independent experiments. *$P < 0.05$; **$P < 0.01$; ***$P < 0.001$.

C    HEK293 was transfected for 48 h with siRNA against ATM (siATM), GSNOR (siGSNOR), or control siRNA (scramble, siScr). Afterward, they were treated for 8 h with 100 μM H$_2$O$_2$ and mitophagy was assessed by Western blot of different mitochondrial complex subunits [i.e., NDUFB8 (complex I), SDHB (complex II), and MTCO2 (complex IV)]. Basal and phospho-ATM and GSNOR were used to check the efficiency of siRNA-mediated knockdown. Vinculin was used as loading control.

D    Densitometry of mitochondrial protein immunoreactive bands of panel C (normalized to Vinculin) is indicated as H$_2$O$_2$-treated versus untreated cells (CTR) and expressed as arbitrary units. Values shown represent the means $\pm$ SD of $n = 3$ independent experiments. *$P < 0.05$; **$P < 0.01$; ***$P < 0.001$; n.s., not significant.

E    In the same experimental settings, mitophagy was also assessed by RT–qPCR relative quantitation of D-loop (selected as measure of mtDNA) normalized to genomic actin (gActin). Results shown are the means $\pm$ SD of $n = 8$ experiments ***$P < 0.001$; n.s., not significant.

F    U2OS cells were depleted of endogenous ATM by repeated transfections with shRNA and induced, by doxycycline incubation, to express ATM$^{WT}$, ATM$^{2RA}$, or ATM$^{CL}$ mutant. Where indicated, cells were further transfected with a GSNOR-coding vector and then treated for 4 h with 100 μM H$_2$O$_2$. Mitophagy was assessed by RT–qPCR relative quantitation of D-loop normalized to genomic actin (gActin). Results shown are the means $\pm$ SD of $n = 6$ experiments. *$P < 0.05$; ***$P < 0.001$.

G, H    In the same experimental settings, mitophagy was also assessed at 8 h by fluorescence microscopy analyses upon incubation with chloroquine (CLQ) to enhance differences in mitophagy. Anti-TOM20 (red) was used to visualize mitochondria; anti-LC3 (green) was used to identify autophagosomes. Percentage of mitochondria merging with LC3-positive puncta calculated by Fiji analysis software using the open-source plugin ComDet v. 0.3.7. Values are expressed as % of mitochondria (TOM20$^+$ particles) co-localizing with LC3/cell and graphed as boxes (25$^{th}$-75$^{th}$ interquartile range) and whiskers (minimum to maximum showing all points), with central bands representing the median of $n \geq 7$ different cells analyzed. *$P < 0.05$; **$P < 0.01$; ***$P < 0.001$.

Source data are available online for this figure.

be physiologically exposed to concomitant H$_2$O$_2$ and NO bursts as downstream effects of receptor engagements (Jackson et al, 2004; Wheeler & DeFranco, 2012; Belikov et al, 2015) and NOS induction (Vig et al, 2004; Niedbala et al, 2006; Saini et al, 2014; Bogdan, 2015), respectively, which is fundamental for immune cell development, proliferation, differentiation, and death. On the basis of the results so far obtained, we hypothesized that ATM/GSNOR axis could exert a protective role during T-cell activation. This would provide the molecular explanation of the evidence that A-T patients (as well as $Atm^{-/-}$ mice) show a reduced number of CD4-single positive (CD4$^+$) cells (Nissenkorn & Ben-Zeev, 2015), which, strikingly, exactly phenocopies $Gsnor^{-/-}$ mice lymphopenia (Yang et al, 2010).

To test this hypothesis, we stimulated Jurkat cells, an immortalized T lymphocyte cell line, with ionomycin and phorbol 12-myristate 13-acetate (PMA), and evaluated H$_2$O$_2$ and NO production by cytofluorometrically following DCF and DAF-FM fluorescence, respectively. As already reported by others (Kamiński et al 2010; Williams et al, 1998), we observed that Jurkat stimulation was associated with NO and H$_2$O$_2$ generation (Fig 8A and B). Western blot analyses also indicated that ATM and GSNOR were induced in these conditions, as well as upon H$_2$O$_2$ treatment, which was used as a positive control (Fig 8C). We then verified whether ATM and GSNOR had a protective role during Jurkat stimulation. To this end, we pre-incubated the cells with KU55933 (KU), or with the pharmacological inhibitor of GSNOR (N6022), and cytofluorometrically analyzed the extent of SubG1 cells upon staining with propidium iodide (PI). Both ATM and GSNOR inhibition exacerbated the detrimental effects of H$_2$O$_2$ and PMA/ionomycin co-treatment (Fig 8D).

To provide further evidence about the biological relevance of our results, we isolated CD4$^+$ T-cell population from human blood (Appendix Fig S5). Purified T cells were activated by incubations with a mix of anti-CD3, anti-CD28 and anti-CD49d; afterward, H$_2$O$_2$ and NO were cytofluorometrically evaluated following DCF and DAF-FM fluorescence. As previously reported (Jackson et al, 2004; Vig et al, 2004; Bogdan, 2015), also in our experimental setting, T-cell stimulation resulted in an increased

production of H$_2$O$_2$ and NO (Fig 8E and F). Then, we pharmacologically inhibited ATM or GSNOR and evaluated cell death extent upon stimulation. Cytofluorometric analyses showed a significant increase of dead (AnV$^+$/PI$^+$) cells (Fig 8G) and a reduced rate of T-cell blast differentiation (Fig 8H) substantiating that the antioxidant/protective role of ATM/GSNOR axis is required in T-cell activation. Actually, this phenomenon was even more pronounced upon ATM inhibition, reasonably due to fact that ATM plays multiple roles in cell homeostasis, the most important of which is to guard any possible defects/damage in DNA, such as those occurring when cells actively replicate. By definition, blast activation implies T cells enter rapid replication phase. Therefore, it is likely that ATM inhibition does not only affect GSNOR-mediated potentiation of mitophagy, but rather impact broadly T-cell differentiation process.

## Discussion

In this work, we provided the first evidence arguing for GSNOR being modulated at a translational level in cells experiencing oxidative stress. Our results support the hypothesis that GSNOR protects cell viability by promoting mitochondrial removal via Parkin-mediated mitophagy. Such an effect, although dispensable under mild H$_2$O$_2$ challenge, becomes essential for cell subjected to severe oxidative stress or to a double oxidative hit, e.g. when H$_2$O$_2$ burst is coupled to concomitant NO fluxes. This condition is frequently faced by some cell types, e.g., macrophages, B and T lymphocytes during immune response, in which, as an effect of infection, or upon tumor cell recognition, iNOS and NADPH oxidases are rapidly and contextually activated. Therefore, by modulating cell response to NO and H$_2$O$_2$, GSNOR reasonably plays a role in correct lymphocyte activation and, generally, in immunocompetence. This hypothesis finds support in the results here obtained in Jurkat and CD4$^+$ T cells and might provide a rationale for $Gsnor$-null mice lymphopenia, which is, indeed, associated with a decreased number of both T and B cells and with an increased rate of apoptosis of CD4$^+$ thymocytes (Yang et al, 2010).

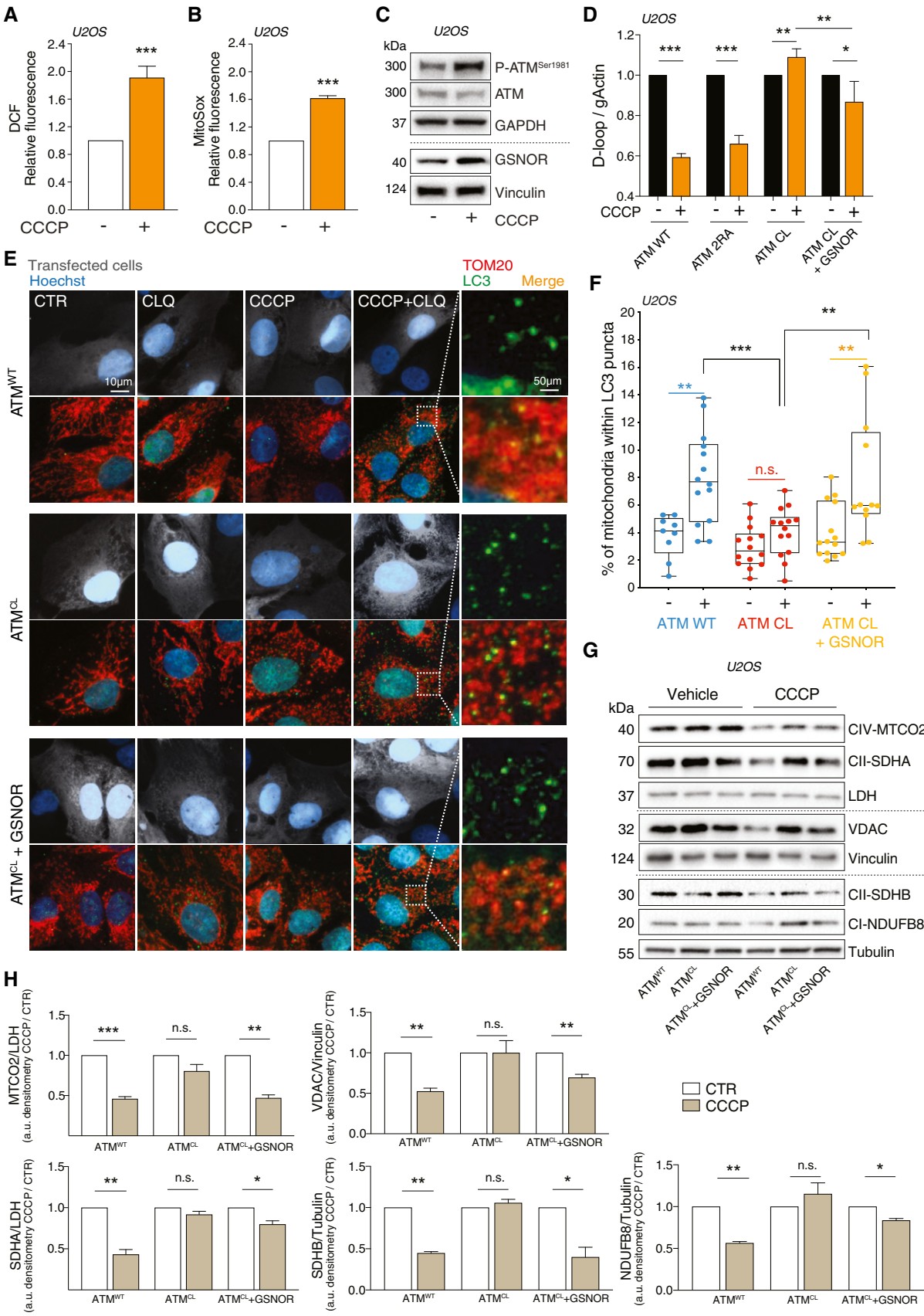

**Figure 6.**

**Figure 6. ATM/GSNOR axis drives mitophagy upon CCCP treatment.**

A, B  ATM^WT U2OS cells were treated with 10 μM CCCP for 8 h. After treatment, cells were incubated with 5 μM 2',7'-H$_2$DCF-DA (A) or MitoSox (B) to evaluate the production of H$_2$O$_2$ or mitochondrial superoxide, respectively. Values are shown as units of DCF or MitoSox fluorescence relative to untreated cells (arbitrarily set as 1) and represent the means ± SD of $n$ = 3 independent experiments. ***$P$ < 0.001.

C  In the same experimental settings, basal and phospho-active forms of ATM and GSNOR were assessed by Western blot. Vinculin and GAPDH were used as loading controls.

D–H  U2OS cells were depleted of endogenous ATM by repeated transfections with shRNA and induced, by doxycycline incubation, to express ATM^WT, ATM^2RA, or ATM^CL mutant. Where indicated, cells were further transfected with a GSNOR-coding vector and then treated for 2 h with 10 μM CCCP. Mitophagy was evaluated by: (D) RT–qPCR relative quantitation of D-loop normalized to genomic actin (gActin). Results shown are the means ± SD of $n$ = 4 experiments. *$P$ < 0.05; **$P$ < 0.01; ***$P$ < 0.001. (E, F) fluorescence microscopy analyses upon incubation with chloroquine (CLQ) to enhance differences in mitophagy. Anti-TOM20 (red) was used to visualize mitochondria; anti-LC3 (green) was used to identify autophagosomes. Percentage of mitochondria merging with LC3-positive puncta calculated by Fiji analysis software using the open-source plugin ComDet v. 0.3.7. Values are expressed as % of mitochondria (TOM20$^+$ particles) co-localizing with LC3/cell and graphed as boxes (25th-75th interquartile range) and whiskers (minimum to maximum showing all points), with central bands representing the median of $n$ ≥ 9 different cells. **$P$ < 0.01; ***$P$ < 0.001; n.s., not significant. (G, H) Western blot of different subunits of mitochondrial proteins, i.e., NDUFB8 (complex I), SDHA and SDHB (complex II), MTCO2 (complex IV) and voltage-dependent anion channel (VDAC). Tubulin, LDH and Vinculin were used as loading controls. Densitometry of mitochondrial protein immunoreactive bands and expressed as arbitrary units. Values shown represent the means ± SD of $n$ = 3 independent experiments. *$P$ < 0.05; **$P$ < 0.01; ***$P$ < 0.001; n.s., not significant.

Source data are available online for this figure.

Our results also demonstrate that GSNOR increase upon H$_2$O$_2$ challenge is an event induced downstream of a signaling axis triggered by the redox activation of ATM, which involves CHK2 and p53 as intermediate players. It is recently emerging that, besides transcriptional regulation, p53 activation is also associated with changes in translational efficiency of a set of mRNAs activated upon stress (Starck et al, 2016; Andrysik et al, 2017; Liang et al, 2018) and required for cell survival, e.g., the uORF-containing genes (Zaccara et al, 2014; Marcel et al, 2018). Work is still in progress in our laboratory to identify the exact mechanism through which GSNOR mRNA is actively translated upon p53 phospho-activation. However, the evidence that 5' UTR of GSNOR mRNA contains two uORF regions (Kwon et al, 2001) let us speculate that it might be target of p53.

Whatever is the molecular mechanism responsible for the enhancement of GSNOR translation, the evidence for a functional relationship between ATM and GSNOR argues for this crosstalk being deeply implicated in human pathophysiology. In support to this assumption, we should consider that one of the hallmarks of ataxia telangiectasia (A-T) is immune deficiency (Nowak-Wegrzyn et al, 2004). This condition phenocopies $Gsnor^{-/-}$ mice lymphopenia and is in line with results obtained in Jurkat and CD4$^+$ T cells, in which pharmacological inhibition of ATM is associated with cell death and reduction of proliferation upon stimulation. Along this line of reasoning, it has been published that GSNOR deficiency brings about excessive S-nitrosylation of proteins regulating mitochondrial dynamics and removal by mitophagy, i.e., Drp1 and Parkin (Rizza et al, 2018). Such a condition results in highly fragmented dysfunctional mitochondria which are not properly degraded, thus implicating GSNOR in cell senescence and mammalian longevity (Rizza & Filomeni, 2018). Remarkably, mitochondrial abnormalities in A-T lymphocytes were already proposed being caused by deficit in mitophagy (Valentin-Vega et al, 2012). It has also been recently observed that cells stably expressing the redox-insensitive mutant of ATM (ATM^CL) correctly repair DNA damage, but show mitochondrial dysfunction, autophagy defects, and early signs of senescence upon oxidative insults (Lee et al, 2018). Here, we showed that ectopic expression of GSNOR is able to complement mitophagy impairments in ATM^CL cells, suggesting that GSNOR might act as downstream effector of ATM in response to oxidative stimuli. This hypothesis is supported by recent findings arguing for ATM redox activation being indispensable for selective removal of peroxisomes by autophagy (the so-called pexophagy), which is triggered to prevent excessive H$_2$O$_2$ production from these organelles (Zhang et al, 2015). In line

**Figure 7. Redox activation of ATM/GSNOR axis is required to counteract nitroxidative stress-induced cell death.**

A  Scheme of the protocol designed for the combined treatment of HEK293 cells with H$_2$O$_2$ and DPTA.

B  HEK293 cells overexpressing the wild-type form of GSNOR (GSNOR^wt) or an empty vector (Empty) were subjected to combined treatment (200 μM H$_2$O$_2$ + 400 μM DPTA) in the presence or absence of the ATM inhibitor KU55933 (KU). Analysis of dead cells was performed with Trypan blue exclusion assay. Western blot analysis of GSNOR is shown as inset in the graph to substantiate transfection efficiency. Data, shown as fold change of dead cells relative to untreated cells (arbitrarily set to 1), represent the mean count ± SEM of $n$ = 4 experiments done in duplicate *$P$ < 0.05; with respect to Empty cells.

C  Representative optic microscopy image of GSNOR^wt and empty cells upon 24-h treatment with H$_2$O$_2$ + DPTA in the presence KU.

D  Western blot analysis of caspase3 and PARP1 in GSNOR^wt and empty cells treated as in panel C. Tubulin was used as loading control.

E  HEK293 cells overexpressing the wild-type form of GSNOR (GSNOR^wt) or an empty vector (Empty) were transfected with siRNA against Pakin (siParkin) or control siRNA (scramble, siScr). Afterward, they were subjected to combined treatment (200 μM H$_2$O$_2$ + 400 μM DPTA) and viability assessed by Alamar blue (AB) fluorescent assay. Data, shown as fold change of viable cells, refer to AB fluorescence (relative to untreated cells, arbitrarily set to 1) and represent the means ± SD of $n$ = 6 independent experiments. **$P$ < 0.01; n.s., not significant.

F  Western blot of Gsnor was assessed in mouse adult fibroblasts (MAFs), obtained from wild-type (WT) and Gsnor-null (KO) mice treated with 200 or 400 μM H$_2$O$_2$ for 24 h.

G, H  WT and Gsnor-null MAFs were subjected to treatment with 200 μM H$_2$O$_2$, or 400 μM DPTA, or a combination of both. Cell viability was evaluated by LIVE/DEAD assay. Scale bar = 50 μm. Data, shown as % of dead (red) cells, represent the mean count ± SD of $n$ = 3 different fields of three independent experiments. *$P$ < 0.05; n.s., not significant with respect to WT MAFs.

Source data are available online for this figure.

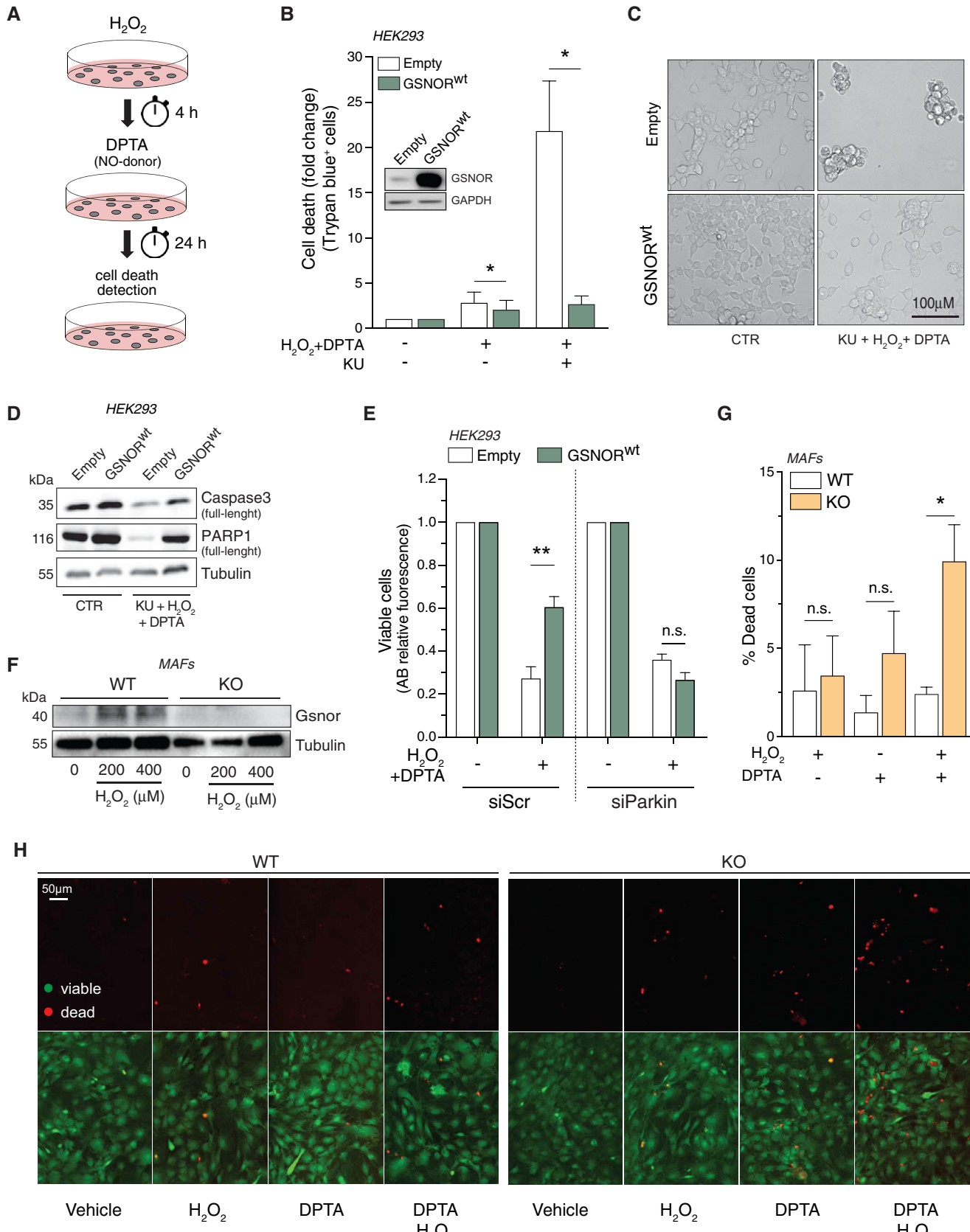

**Figure 7.**

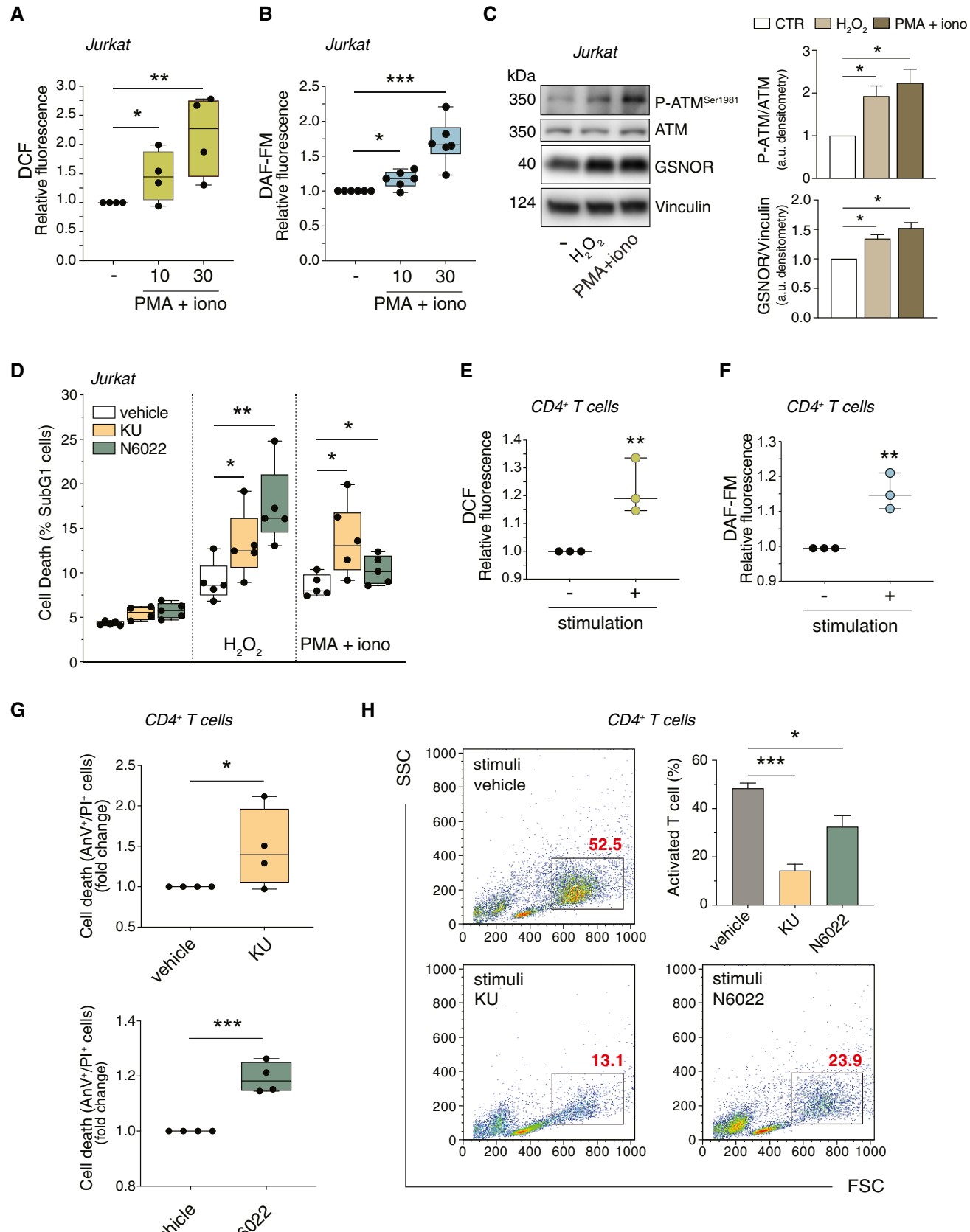

**Figure 8.**

**Figure 8.  Role of ATM and GSNOR in T-cell activation.**

A, B   Jurkat cells were treated for 10 and 30 min with PMA (200 ng/ml) and ionomycin (*iono*; 300 ng/ml). After treatment, cells were incubated with 5 μM 2',7'-H$_2$DCF-DA (A) or DAF-FM-DA (B) to evaluate the production of H$_2$O$_2$ or NO, respectively. Values are expressed as units of DCF or DAF-FM fluorescence relative to untreated cells (arbitrarily set as 1) and graphed as boxes (25$^{th}$-75$^{th}$ interquartile range) and whiskers (minimum to maximum showing all points), with central bands representing the median of $n = 4$ (A) and $n = 6$ (B) independent experiments. *$P < 0.01$; **$P < 0.001$; ***$P < 0.001$.

C   (*left*) Jurkat cells were treated for 24 h with H$_2$O$_2$ (50 μM) or PMA/ionomycin (200 + 300 ng/ml). Basal and phospho-active form of ATM, and GSNOR were assessed by Western blot. Vinculin was used as loading control. (*right*) Phospho:basal level ratios of ATM along with densitometry of GSNOR immunoreactive bands are expressed as arbitrary units. Values shown represent the means ± SD of $n = 3$ independent experiments. *$P < 0.05$.

D   Jurkat cells were treated for 24 h with H$_2$O$_2$ (50 μM) or PMA/ionomycin (200 + 300 ng/ml), in the presence or absence of ATM inhibitor (KU55933; 5 μM) or GSNOR inhibitor (N6022; 25 μM). Cell death was assessed cytofluorimetrically upon staining with propidium iodide (PI). Values are expressed as % of sub-G1 population of PI-stained cells and graphed as boxes (25$^{th}$-75$^{th}$ interquartile range) and whiskers (minimum to maximum showing all points), with central bands representing the median of $n = 5$ independent experiments. *$P < 0.05$; **$P < 0.01$.

E, F   CD4$^+$ T cells were incubated for 30 min with anti-CD3, anti-CD28 and anti-CD49d (*stimulation*). After stimulation, cells were incubated with 5 μM 2',7'-H$_2$DCF-DA (E) or DAF-FM-DA (F) to fluorometrically evaluate the production of H$_2$O$_2$ or NO, respectively. Values are shown as units of DCF or DAF-FM fluorescence relative to non-stimulated cells (arbitrarily set as 1). Values are shown as fold change and represent the median plus range with all the experimental points of $n = 3$ independent experiments. **$P < 0.01$.

G   CD4$^+$ T cells were stimulated for 96 h with anti-CD3, anti-CD28 and anti-CD49d in the presence or absence of ATM inhibitor (KU, *upper panel*) or GSNOR inhibitor (N6022, *bottom panel*). Cell death was assessed cytofluorimetrically upon staining with Annexin V (AnV) and PI. Values are expressed as fold change of AnV$^+$/PI$^+$ cells relative to control (CD4$^+$ without inhibitor, *vehicle*, arbitrarily set to 1) and graphed as boxes (25$^{th}$-75$^{th}$ interquartile range) and whiskers (minimum to maximum showing all points), with central bands representing the median of $n = 4$ independent experiments. *$P < 0.05$; ***$P < 0.001$.

H   In the same experimental settings, populations of stimulated (proliferating) CD4$^+$ T cells were identified cytofluorometrically and included in rectangles. Values (as % of total population) are shown in red in each representative plot identify and summed up in a graph as the means ± SD of $n = 3$ independent experiments. *$P < 0.05$; ***$P < 0.001$.

Source data are available online for this figure.

with these observations, a very recent paper has provided evidence that the activation of ATM/CHK2 signaling pathway is required, upon prolonged nutrient or oxygen deprivation, to attenuate ROS production *via* the induction of Beclin1-mediated autophagy (Guo *et al*, 2020). Our findings complement all these data and contribute to provide a comprehensive explanation why ATM deficiency has been often related to oxidative stress conditions. In addition, they suggest that: i) a number of disturbances arising from ATM mutation or deficiency (e.g., A-T and different cancer types) could partly result from deficits in GSNOR-mediated mitophagy; ii) increasing cellular denitrosylating capacity can represent a good tool to complement ATM in such diseases. In agreement with these assumptions, it has been reported that *Atm*$^{-/-}$ mice show a decreased thiol redox activity, which is restored by administrations with the antioxidant and denitrosylating molecule *N*-acetyl cysteine (NAC; Yan *et al*, 2001).

## Materials and Methods

### Cell culture

*Cell lines*—HEK293, HCT116, SAOS, U2OS, HeLa, and BJ-hTERT cells were grown in Dulbecco's modified Eagle's medium (DMEM). Jurkat cells were grown in RPMI-1640. Both DMEM and RPMI-1640 were supplemented with 10% fetal bovine serum (FBS), 100 U/ml penicillin, and 100 μg/ml streptomycin (Thermo Fisher Scientific). All cells are form ATCC and were maintained in a humidified 5% CO$_2$, 37°C incubator, unless they were subjected to hypoxia conditions, achieved by culturing HEK293 cells for 4 or 8 h in a hypoxia chamber (*in vivo* RUSKINN, Baker) set at $pO_2$ 1%.

U2OS Flip-In T-REx cells containing wild-type (WT), 2RA, CL forms of ATM alleles, were cultured in Dulbecco's modified Eagle's medium (DMEM; Thermo Fisher Scientific) supplemented with 10% FBS (Thermo Fisher Scientific) containing blasticidin (15 mg/ml; Sigma-

Aldrich), penicillin-streptomycin (100 U/ml; Thermo Fisher Scientific), and hygromycin (200 mg/ml; Sigma-Aldrich). Depletion of endogenous ATM was performed by transfecting shRNA against ATM (see transfection section) every 48 h for three consecutive times. Where indicated, the last transfection was conducted along with a GSNOR expressing plasmid, to concomitantly overexpress GSNOR. The last three days of experiments, doxycycline (1 mg/ml; Sigma-Aldrich) was added to the medium to induce expression of ATM$^{WT}$, ATM$^{CL}$ or ATM$^{2RA}$ mutants.

*Mouse adult fibroblasts (MAFs)* were obtained from *wild-type* (WT) and Gsnor-null (KO) mouse ears. Briefly, ear explants were minced and incubated with basal DMEM-containing collagenase (1 mg/ml) for 1 h at 37°C, 5% CO$_2$. Then, DMEM containing 10% FBS was added and pellet centrifuged at 500 $g$ to remove collagenase traces. Ears' extracts were then laid down in tissue culture dishes in complete DMEM (10% FBS and antibiotics) to let the cells spill out from tissue. After 7–10 days, tissues debris was removed and cell let grown in DMEM supplemented with 10% FBS, 100 U/ml penicillin, and 100 μg/ml streptomycin.

*CD4$^+$ T lymphocytes*—Peripheral blood mononuclear cells (PBMC) from healthy donors were isolated by Ficoll density gradient centrifugation, and CD4$^+$ T lymphocytes were sorted using "CD4$^+$ T Cell Isolation Kit, human" (by Miltenyi Biotec), according to manufacturer's instructions. Purified cells were cultured in complete medium (RPMI 1640 supplemented with 10% fetal bovine serum, 5 mM glutamine, and 5 μg/ml gentamicin), at the concentration of 10$^6$ cells/ml, and stimulated with 1 μg/ml of anti-CD28, anti-CD3, and anti-CD49d monoclonal antibodies (Miltenyi Biotec). Cell purification was assessed by flow cytometry (FACSCelesta, BD) by CD4 and CD3 staining being in the range of 95% (Appendix Fig S5). Finally, T-cell blast differentiation was assessed by flow cytometric analysis of FSC vs. SSC, as a simple and suitable method for detecting T-cell activation (Böhmer *et al*, 2011). All procedures described herein were authorized by the Ethics committee of University of Rome "Tor Vergata" (# R.S. 17/20).

## Reagents

$H_2O_2$, KU55933, AZD7762, N6022, PMA, ionomycin, pifithrin-α, CCCP, cycloheximide, trigonelline, dipropylenetriamine (DPTA) NONOate, hydroxyurea, neocarzinostatin, trypan blue, antimycin A, oligomycin A, and salts used for buffers were from Sigma-Aldrich. All the compounds were used in a range of concentrations that did not *per se* induce any significant modulations of GSNOR levels (Appendix Fig S6), i.e., KU55933, 5 μM; AZD7762, 20 nM; and pifithrin-α, 20 μM. Other concentration used are as follows: $H_2O_2$, 100, 200 or 400 μM; cycloheximide, 30 μM; trigonelline, 2.5 μM; neocarzinostatin, 0.5 μg/ml; DPTA, 400 μM; HU, 2 mM; CCCP, 5 or 10 μM; N6022, 25 μM; PMA, 200 ng/ml; ionomycin, 300 ng/ml; antimycin A, 1 μM; and Oligomycin A, 1 μM, unless otherwise indicated.

## Transfections

Transient knocking down of ATM, CHK1, CHK2, GSNOR, Parkin, and p53 was performed by transfecting cells with commercially available endoribonuclease-prepared siRNA pool (esiRNA, Sigma-Aldrich), while controls were transfected with a scramble siRNA duplex (siScr), which does not present homology with any other human mRNAs. siRNAs were transfected using Lipofectamine 3000 (Thermo Fisher Scientific), according to manufacturer's instructions. Overexpression of GSNOR and $p53^{wt}$ was performed using PEI (Tebu-bio).

shRNA used to stably knockdown ATM was designed in our laboratory and synthesized by TAG Copenhagen.

| Top | 5'-TGCTG**CTTTTATGAGCACCATCTTCA**GTTTTGGCCACTGACTGAC**TGAAGATTGCTCATAAAAG**-3' |
|---|---|
| Bottom | 5'-CCTG**CTTTTATGAGCAATCTTCA**GTCAGTCAGTGGCCAAAACT**TGAAGATGGTGCTCATAAAAGC**-3' |

Constructs were cloned in pcDNA6.2-GW/EmGFP-miR vector using the BLOCK-iT™ Pol II miR RNAi Expression Vector Kit with EmGFP (Thermo Fisher Scientific), in according to manufacturer's instructions, and were transfected using Genejuice (Merck).

## Analysis of cell death and viability

*HEK293*—Cells were treated with 200 μM $H_2O_2$ for 4 h; then, the medium was replaced with a fresh one containing 400 μM DPTA and cells maintained for 24 h. Afterward, cells were detached and cell death was evaluated by direct cell count upon Trypan blue exclusion assay. Alternatively, cell viability was quantified by reading the fluorescence emission at 590 nm after 2-h incubation with AlamarBlue® Reagent (Thermo Fisher Scientific) with a Victor X4 (PerkinElmer) plate reader.

*MAFs*—Cells were treated with 200 μM $H_2O_2$ for 4 h; then, the medium was replaced with a fresh one containing 400 μM DPTA and cells there maintained for further 24 h. Next, cells were stained with LIVE/DEAD Cell Imaging Kit (488/570; Thermo Fisher Scientific) and cell death evaluated by fluorescence microscopy.

*Jurkat cells*—Cell was treated with 50 μM, $H_2O_2$ or, alternatively, Ionomycin (300 ng/ml) and PMA (200 ng/ml) in the presence or absence of 5 μM KU55933, or 25 μM N6022. After 24 h of treatment, cell death was evaluated by flow cytometry (FACSCalibur, BD) upon propidium iodide staining, measuring the percentage of sub-G1 cell fraction.

*CD4+ T lymphocytes*—CD4+ T cells were stimulated for 96 h with 1 μg/ml anti-CD3, anti-CD28, and anti-CD49d in the presence or absence of 10 μM KU55933 or 25 μM N6022. Cells were then collected and double-stained with annexin V-FITC (anV) and propidium iodide (PI) as recommended by the supplier (Molecular Probes). Apoptotic cells were measured by flow cytometry (FACSCalibur, BD).

## $H_2O_2$, mitochondrial superoxide, and NO evaluation

*HEK293 and U2OS cells*—Soon after treatments, cells were incubated with 5 μM MitoSox or 2',7'-dihydrodichlorofluorescein diacetate ($H_2DCFDA$) at 37°C for the detection of mitochondrial superoxide or $H_2O_2$, respectively. Stained cells were washed twice with cold PBS, collected, and analyzed by flow cytometry (FACS Verse, BD-biosciences).

*CD4+ T cells*—T cells isolated from healthy donors were loaded for 30 min at 37°C with 2 μM of $H_2DCFDA$ or 4-amino-5-methylamino-2',7'-difluorofluorescein diacetate (DAF-FM-DA; Molecular Probes) for the detection of $H_2O_2$ and NO, respectively. Then, cells were stimulated for 15 min with 1 μg/ml of anti-CD28, anti-CD3 and anti-CD49d monoclonal antibodies (Miltenyi Biotec). ROS and NO generation were evaluated by fluorometric assay by setting the wavelength of excitation/emission at 488 nm/530 nm for $H_2O_2$ and 480 nm/520 nm for NO. Fluorescence has been evaluated by a Varioskan LUX Multimode Microplate Reader (Thermo Fisher Scientific).

*Jurkat*—Jurkat cells were loaded with 4 μM $H_2DCFDA$ or 2 μM DAF-FM as above described. Then, cells were washed and stimulated with Ionomycin (300 ng/ml) and PMA (200 ng/ml) for 10 and 30 min at 37°C. $H_2O_2$ and NO generation was assessed by flow cytometry (FACSCelesta, BD-biosciences).

## Western blot analyses

Total protein lysates were obtained by rupturing cells in RIPA buffer (50 mM Tris–HCl, pH 8, 150 mM NaCl, 1% NP-40, 0.5% sodium deoxycholate, 0.1% SDS, 10 mM NaF, 1 mM sodium orthovanadate) and protease inhibitor cocktail (Roche Applied Science) followed by centrifugation at 22,300 g for 20 min at 4°C. Twenty μg protein extracts were then electrophoresed by SDS–PAGE and blotted onto nitrocellulose membrane (GE Healthcare). Alternatively, to detect GSNOR protein, protein lysates were obtained by rupturing cells with 30 min of incubation on ice in lysis buffer (10 mM HEPES pH 7.5, 150 mM NaCl, 10 mM NaF, 1 mM sodium orthovanadate) and protease inhibitor cocktail followed by sonication and centrifugation at 22,300 g for 20 min at 4°C. Primary antibodies used are as follows: anti-GAPDH (sc-47724), anti-LDH (sc-33781), anti-ATM (sc-23921), anti-p53 (sc-126), anti-lamin A/C (sc-20681), anti-ATR (sc-1887), anti-CHK1 (G4), anti-GSNOR (sc-293460), anti-TOM20 (sc-11415), anti-VDAC1 (sc-8828; Santa Cruz Biotechnology); anti-tubulin (T9026; Sigma-Aldrich); anti-GSNOR (ABC383), and anti-CHK2 (05-649; Merck Millipore); anti-phospho-ATM-Ser[1981] (10H11.E12), anti-phospho-CHK2-Thr[68] (2661), anti-phospho-CHK1-Ser[317] (2334), anti-phospho-ATR-Ser[428] (2853); anti-phospho-CHK1-Ser[345] (2341); anti-phospho-p53-Ser[15] (9284); anti-Nrf2 (12721); anti-VDAC (4866; Cell Signaling Technology); anti-Vinculin (18058; Abcam); anti-γH2A.X (3F); anti-HIF-1α (GTX127309; GeneTex);

anti-Hsp90 (AC-88; Stressgen—Enzo Life Sciences); and anti-SDHA (ab14715), anti-MTCO2, anti-SDHB, and anti-NDUFB8 (from total OXPHOS antibody cocktail, ab110413; Abcam). The specific protein complex, formed upon incubation with specific secondary antibodies (Bio-Rad Laboratories), was identified using a LAS-3000 Imaging System (Fujifilm) or Chemidoc Imaging System XRS+ (Bio-Rad), after incubation with the ECL detection system (LiteABlot Turbo, EuroClone). Images were adjusted for brightness and contrast by Fiji (Schindelin *et al*, 2012) analysis software.

### Evaluation of cellular protein synthesis

HEK293 cells were treated for 10 min with 1 μM puromycin and lysed in Ripa buffer supplemented with protease and phosphatase inhibitors after 50 min of chasing. Cellular protein content was quantified using the Pierce BCA protein assay kit (Thermo). Sixty μg protein samples were resolved on 4–12% NuPAGE gel (Life Technologies) and transferred to a nitrocellulose membrane (Bio-Rad). Western blot analyses with a mouse IgG2a monoclonal anti-puromycin antibody (Merck Millipore, 1:5,000) and proteins visualized using an ECL system (Bio-Rad). Pretreatment with the translation inhibitor cycloheximide (CHX) were performed as negative controls as it blocks puromycin incorporation.

### Detection of S-nitrosylated proteins (PSNOs)

Protein *S*-nitrosylation was evaluated by biotin-switch assay as previously described (Montagna *et al*, 2014). Briefly, cells were homogenized in HEN buffer (25 mM HEPES, 50 mM NaCl, 0.1 mM EDTA, 1% NP-40, protease inhibitors, pH 7.4). Free cysteine residues were blocked with *S*-methyl methanethiosulfonate (MMTS, Thermo Fisher Scientific), diluted 1:10 with a solution containing SDS (2.5% final concentration), and incubated for 15 min at 50°C. Proteins were then precipitated with cold acetone for 20 min at −20°C, collected by centrifugation, resuspended in HEN buffer with 1% SDS, and incubated with biotin-HPDP (2.5 mg/ml) in the presence or absence of 20 mM sodium ascorbate. After incubation with the HRP-conjugated streptavidin (Merck Millipore), biotinylated proteins were revealed using the ECL detection system.

### Cell fractionation

Fresh cellular pellets were gently lysed in *nucleus* buffer (10 mM HEPES, pH 7.9, 10 mM KCl, 1.5 mM MgCl₂, 0.5 mM DTT) containing protease and phosphatase inhibitor cocktails. After 10 min on ice, 0.1% IGEPAL was added and cells incubated for further 20 min. Cytosolic fraction was obtained after centrifugation at 12,000 *g* for 30 s at 4°C. Nuclear pellet was washed twice with cold PBS, lysed in RIPA, sonicated, and centrifuged at 12,000 *g* for 10 min at 4°C.

### Polysomal fractionation

Polysomal fractionation was performed as previously described (Gandin *et al*, 2014); briefly, prior collection, cells were treated with 100 μg/ml cycloheximide, washed, and collected in ice-cold PBS supplemented with 100 μg/ml cycloheximide. Next, cells were centrifuged at 5,000 rpm for 5 min at 4°C and lysed in ice for 30 min in hypotonic buffer [(5 mM Tris–HCl (pH 7.5), 2.5 mM MgCl₂, 1.5 mM KCl, and 1× protease inhibitor cocktail (EDTA-free)]. Afterward, 5 μl of 10 mg/ml cycloheximide, 1 μl of 1 M DTT, and 100 units of RNAse inhibitor were added and lysates vortexed for 5 s followed by addition of 25 μl of 10% Triton X-100 (final concentration 0.5%) and 25 μl of 10% sodium deoxycholate (final concentration 0.5%). Lysates were centrifuged at 16,000 *g* for 10 min at 4°C, and supernatants ware transferred and normalized accordingly to OD 260 nm. Ten percent of the sample was kept as input, and the rest of the lysate was separated on 5–50% sucrose linear density gradient by centrifuging 222,228 *g* for 2 h at 4°C. The polysomal fractions were monitored and collected using a gradient fractionation system (Isco).

For further analysis, total RNA was extracted from input and heavy polysomal fractions (200 μl from each of the last six fractions were pooled together) were extracted per each samples using TRIzol LS (Thermo Fisher Scientific) accordingly to manufacturer protocol. qPCR experiments were performed as described in the following section.

### Real-Time PCR (RT–qPCR)

Cells were homogenized in TRI Reagent (Sigma-Aldrich), and RNA was extracted in accordance with manufacturer protocol. Total RNA was solubilized in RNase-free water, and first-strand cDNA was generated starting from 1 μg of total RNA using the GoScript Reverse Transcription System (Promega). In order to hybridize to unique regions of the appropriate gene sequence, specific sets of primer pairs were designed and tested with primerBLAST (NCBI, see list below). RT–qPCR was performed using the iTAQ universal SYBR Green Supermix (Bio-Rad Laboratories) on a ViiA 7 Real-Time PCR System (Applied Biosystems). Data were analyzed by the ViiA™ 7 Software using the second-derivative maximum method. The fold changes in mRNA levels were determined relative to a control after normalizing to the internal standard actin.

Primers used are listed below:

| Gene | Forward primer | Reverse primer |
|---|---|---|
| hActin | 5'-GGCCGAGGACTTTGATTGCA-3' | 5'-GGGACTTCCTGTAACAACGCA-3' |
| hGSNOR | 5'-CATTGCCACTGCGGTTTGCCAC-3' | 5'-AGTGTCACCCGCCTTCAGCTTAGT-3' |
| hHMOX-1 | 5'-CACAGCCCGACAGCATGCCC-3' | 5'-GCCTTCTCTGGACACCTGACCCT-3' |
| hGCL | 5'-CGCACAGCGAGGAGCTTCGG-3' | 5'-CTCCACTGCATGGGACATGGTGC-3' |
| hH3A | 5'-AAGCAGACTGCCCGCAAAT-3' | 5'-GGCCTGTAACGATGAGGTTTC-3' |

### Mitochondrial DNA relative quantitation (D-Loop)

Total DNA was extracted from cells by E.Z.N.A.® Tissue DNA Kit (Omega Bio-Tek) in accordance with manufacturer protocol. DNA content was measured with a NanoDrop™ 2000 Spectrophotometer (Thermo Fisher scientific), and 5 μg used for mitochondrial DNA content measurement by RT–qPCR. The relative quantitation of mitochondrial D-Loop region was normalized to genomic Actin (gActin), and it was achieved by using iTAQ universal SYBR Green Supermix (Bio-Rad Laboratories) and a ViiA 7 Real-Time PCR

System (Applied Biosystems). All reactions were run as triplicates. Data were analyzed by the ViiA™ 7 Software using the second-derivative maximum method. Primers used were as follows:

| Gene | Forward primer | Reverse primer |
|------|----------------|----------------|
| gActin | *5'-CCCCTGGCGGCCTAAGGACT-3'* | *5'-ACATGCCGGAGCCGTTGTCG-3'* |
| D-Loop | *5'-ACCACCCAAGTATTGACTCACC-3'* | *5'-CCGTACAATATTCATGGTGGCT-3'* |

### Fluorescence microscopy

*γH2A.X*: After 2-h treatment with $H_2O_2$, cells were fixed with 4% paraformaldehyde, incubated with a permeabilization solution (PBS/Triton X-100 0.4% v/v), and blocked for 1 h with a blocking solution (PBS/FBS 10% v/v). Afterward, cells were incubated for 1 h with anti-γH2A.X antibody and for a further one with Alexa Fluor 594 dye-conjugated secondary antibody (Thermo Fisher Scientific) to selectively reveal *foci* of the DNA. Nuclei were stained with Hoechst 33342 (Thermo Fisher Scientific). Images of cells were digitized with a Delta Vision Restoration Microscopy System (Applied Precision) equipped with an Olympus IX70 fluorescence microscope.

*Mitophagy*: Cells were incubated with a permeabilization solution (PBS/Triton X-100 0.2% v/v), blocked for 1 h with a blocking solution (PBS/normal goat serum 5% v/v, FBA 1% v/v), and then incubated over night with anti-LC3 (NanoTools) and anti-TOM20 (Santa Cruz Biotechnology). Cells were then washed with cold PBS and incubated for 1 h with fluorophore-conjugated secondary antibodies (respectively, Alexa Fluor 647 and 568). Nuclei were stained with 1 µg/ml Hoechst 33342 (Thermo Fisher Scientific). Confocal microscopy experiments were performed by using LSM800 microscope (ZEISS) equipped with ZEN imaging software, and fluorescence images were adjusted for brightness, contrast, and color balance by Fiji (Schindelin *et al*, 2012) analysis software. Mitophagy rate was assessed upon 2-h incubation with chloroquine (CLQ, 20 µM) and CCCP (5 or 10 µM) by counting the percentage of mitochondria within LC3-positive puncta only in shATM-transfected cells (expressing EmGFP). 3D projection was achieved by summing the fluorescence signal of the central z-Stacks (four planes, 0.3 µm). At least nine different cells/experimental condition were analyzed by Fiji analysis software using the open-source plugin ComDet v. 0.3.7. The plugin finds and analyzes co-localization of bright intensity spots in images with heterogeneous background. For both TOM20 and LC3 fluorescence channels, the parameters utilized were as follows: particle size ≥ 4 px; intensity threshold = 3. The co-localization was considered positive if the maximum distance between the center of two particles was ≤ 6 px (Appendix Fig S7).

### Total protein assessment

Protein concentration was determined by the method of Lowry *et al* (1951).

### Statistical analysis

Data were from at least three independent experiments, unless otherwise indicated. The results are presented as means ± SD (or SEM where indicated). Statistical evaluation was conducted by using unpaired (or paired for Western blot densitometry) two-tailed Student's *t*-test. Comparisons were considered significant with $P < 0.05$.

## Data availability

The authors confirm that all the data supporting the findings of this study are available within the article and its supplementary materials/source files. This study includes no data deposited in external repositories.

**Expanded View** for this article is available online.

### Acknowledgements
The authors are grateful to Laila Fisher for secretarial work. This work has been supported by grants from the Novo Nordisk Foundation (2018-0052550 to G.F); Danish Cancer Society (KBVU R146-A9414 and R231-A13855 to G.F.; KBVU R146-A9364; and R231-A14034 to F.C.); the Italian Association for Cancer Research, (IG2016-19069 to D.B. and IG2017-20719 to G.F.); Italian Ministry of University and Research (MIUR-JPI-HDHL-NUTRICOG-MiTyrAge and PRIN-2015LZE9944 to D.B); and Italian Ministry of Health (RF-2016-02362022 to D.B.). Chiara Pecorari is recipient of a PhD fellowship from Danish Cancer Research Foundation (Dansk Kraeftforskningsfond, DKF-0-0-532). Giuseppina Claps was supported by the Marie Curie, Campus France Fellowship - PRESTIGE-2017-3-0017. Moreover, laboratory in Copenhagen is part of the Center of Excellence in Autophagy, Recycling and Disease (CARD), funded by the Danish National Research Foundation (DNRF125).

### Author contributions
GF and CC conceptualized the study. CC, SR, MFA, CP, J-HL, PG, NP, GC, BB, TTP, and JSS involved in methodology. CC, SR, CP, GC, PG, NP, and GF investigated the study. TTP, SR, CR, FC, JSS, and GF provided the resources. BB, DB, TTP, MF, CC, SR, and GF reviewed and edited the manuscript. BB, DB, CR, MF, JSS, and GF involved in supervision.

### Conflict of interest
The authors declare that they have no conflict of interest.

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
