## [Review Process File · EMBO Reports]

Redox activation of ATM enhances GSNOR translation to sustain mitophagy and tolerance to oxidative stress

Claudia Cirotti, Salvatore Rizza, Paola Giglio, Noemi Poerio, Maria Francesca Allegra, Giuseppina Claps, Chiara Pecorari, Ji-Hoon Lee, Barbara Benassi, Daniela Barilà, Caroline Robert, Jonathan Stamler, Francesco Cecconi, Maurizio Fraziano, Tanya Paull and Giuseppe Filomeni,
DOI: [10.15252/embr.202050500](https://doi.org/10.15252/embr.202050500)

Corresponding author(s): Giuseppe Filomeni (giuofil@cancer.dk)

Review Timeline:

Submission Date:	24th Mar 20
Editorial Decision:	30th Apr 20
Revision Received:	20th Aug 20
Editorial Decision:	28th Sep 20
Revision Received:	1st Oct 20
Accepted:	14th Oct 20

Transaction Report:

Dear Dr. Filomeni

Thank you for the submission of your research manuscript to our journal. We have now received the full set of referee reports that is copied below.

As you will see, the referees acknowledge that the findings are potentially interesting. However, the referees also indicate that further work will be required to substantiate the findings and to explain some inconsistencies. All three referees note that the CHK2 inhibitor fails to efficiently inhibit CHK2 phosphorylation, yet still has a major impact on GSNOR levels. Moreover, the link between mitophagy and cell death needs to be strengthened and causality should be tested. Moreover, referee 1 points out that the link between ATM/p53 and GSNOR translation has not been explored in further depth and that the regulation by p53 might be indirect. Since the identification of the mechanism by which p53 regulates GSNOR translation might be beyond the scope of this manuscript, it is not essential to address it experimentally but the related conclusions need to be toned down accordingly.

Given these constructive comments, we would like to invite you to revise your manuscript with the understanding that the referee concerns (as detailed above and in their reports) must be fully addressed and their suggestions taken on board. Please address all referee concerns in a complete point-by-point response. Acceptance of the manuscript will depend on a positive outcome of a second round of review. It is EMBO reports policy to allow a single round of revision only and acceptance or rejection of the manuscript will therefore depend on the completeness of your responses included in the next, final version of the manuscript.

We invite you to submit your manuscript within three months of a request for revision. This would be July 30 in your case. Yet, given the current COVID-19 related lockdowns of laboratories, we have extended the revision time for all research manuscripts under our scoping protection to allow for the extra time required to address essential experimental issues. Please contact us to discuss the time needed and the revisions further.

- 1) A data availability section is missing.
- 2) Your manuscript contains error bars based on $n=2$. Please use scatter blots showing the individual datapoints in these cases. The use of statistical tests needs to be justified.

2) individual production quality figure files as .eps, .tif, .jpg (one file per figure).

Please download our Figure Preparation Guidelines (figure preparation pdf) from our Author Guidelines pages

<https://www.embopress.org/page/journal/14693178/authorguide> for more info on how to prepare your figures.

4) a complete author checklist, which you can download from our author guidelines (). Please insert information in the checklist that is also reflected in the manuscript. The completed author checklist will also be part of the RPF.

5) Please note that all corresponding authors are required to supply an ORCID ID for their name upon submission of a revised manuscript (). Please find instructions on how to link your ORCID ID to your account in our manuscript tracking system in our Author guidelines

()

6) We replaced Supplementary Information with Expanded View (EV) Figures and Tables that are collapsible/expandable online. A maximum of 5 EV Figures can be typeset. EV Figures should be cited as 'Figure EV1, Figure EV2" etc... in the text and their respective legends should be included in the main text after the legends of regular figures.

7) All manuscripts require a formal "Data Availability " section (placed after Materials & Method) that follows the model below (see also <
<https://www.embopress.org/page/journal/14693178/authorguide#dataavailability>>).

Please note that the Data Availability Section is restricted to new primary 'omics' data that are part of the study. In case you have no data deposited, please add a statement that no data have been deposited in public databases.

8) We would also encourage you to include the source data for figure panels that show essential data. Numerical data should be provided as individual .xls or .csv files (including a tab describing the data). For blots or microscopy, uncropped images should be submitted (using a zip archive if multiple images need to be supplied for one panel). Additional information on source data and instruction on how to label the files are available .

10) Regarding data quantification:

- Please ensure to specify the name of the statistical test used to generate error bars and P values, the number (n) of independent experiments underlying each data point (not replicate measures of one sample), and the test used to calculate p-values in each figure legend. Discussion of statistical methodology can be reported in the materials and methods section, but figure legends should contain a basic description of n, P and the test applied.

IMPORTANT: Please note that error bars and statistical comparisons may only be applied to data obtained from at least three independent biological replicates. If the data rely on a smaller number of replicates, scatter blots showing individual data points must be used.

- Graphs must include a description of the bars and the error bars (s.d., s.e.m.).

11) As part of the EMBO publication's Transparent Editorial Process, EMBO reports publishes online a Review Process File to accompany accepted manuscripts. This File will be published in conjunction with your paper and will include the referee reports, your point-by-point response and all pertinent correspondence relating to the manuscript.

I look forward to seeing a revised version of your manuscript when it is ready. Please let me know if you have questions or comments regarding the revision.

Yours sincerely

Martina Rembold, PhD
Editor
EMBO reports

Referee #1:

In the manuscript entitled "Redox activation of ATM enhances GSNOR translation to sustain

mitophagy and tolerance to oxidative stress" Cirotti et al. propose the existence of an ATM/ S-nitrosoglutathione reductase (GSNOR) axis. The role of S-nitrosylation in regulating cellular redox balance and survival fate is attracting significant interest. In this manuscript, the authors attempted to explore the intrinsic mechanism on redox stress response by S-nitrosylation factors. They found that ATM activation induced translational upregulation of GSNOR, with a consequent increase in mitophagy to eliminate damaged mitochondria, thereby preventing cell death caused by mild hydrogen peroxide stimulation. The authors point out that the ATM-GSNOR axis could be an oxidative stress sensor to fine-tune cell survival by linking together other quality control systems such as mitophagy. Their findings unveil an interesting role of ATM in the regulation of S-nitrosoglutathione reductase (GSNOR). Together with the well-known ability of GSNOR to regulate lymphocyte development, their finding could potentially contribute to explaining the lymphopenia observed in ataxia telangiectasia patients.

Although the mechanism proposed is interesting and quite novel, due to experimental weaknesses, several improvements are needed prior to eventually granting publication. Specifically,

a. I am concerned about the poor quality of the data, specifically of some of the western blots presented. Most if not all of them need quantification to be able to draw any solid conclusion.

b. In Fig 2, the authors describe the dependency of GSNOR activation (defined as translational regulation in Fig 1) on ATM and CHK2. Yet of the inhibitors tested, KU effectively inhibits ATM phosphorylation, while AZD does not seem to inhibit efficiently CHK2 phosphorylation in Fig 2C, and yet, they both reduce GSNOR. Likewise, in Fig 2E (needs quantification by densitometry), CHK2 targeting siRNAs have very mild effect in silencing CHK2 and therefore in preventing its activation. Altogether, the regulatory effect of ATM on GSNOR seems interesting and plausible but the overall underlying mechanism is not completely clear and the proposed involvement of CHK2 is questionable.

c. The authors further show that NCS treatment does not induce GSNOR expression as well as oxidative stress (H₂O₂). Data shown in western blots should be quantified and accompanied by ROS quantification upon NCS treatment, by flow cytometry (CellRox or similar) to compare the amount of ROS produced by NCS or H₂O₂ treatment and exclude that the induction of DNA damage by NCS in this context generates as much oxidative stress as treatment with H₂O₂. Similarly, data in Fig 3E are misleading, as in the "empty" lane H₂O₂ does not seem as effective as in the other experiments in inducing ATM phosphorylation; nevertheless, GSNOR appears to be effectively induced. In addition, it is not clear why the overexpression of wild type ATM would increase GSNOR expression, making it very hard to evaluate its phosphorylation in the western blots presented (Fig 3E) without proper quantification.

d. The involvement of ATM and p53 in the regulation of GSNOR presented in Fig. 4 and the correlation with the survival advantage provided by this axis in T-cells (Fig 8) is perhaps the most intriguing part of the manuscript but remains poorly explored. Indeed, at the translational level P53 mostly regulates negatively other effectors (i.e. those involved in the cell cycle). The authors here propose that ATM/P53 axis positively regulates at translational level GSNOR. It is likely that GSNOR regulation by p53 is indirect. Several studies have already identified other p53 translational targets (e.g. Zaccara et al, Cell Death Differ., 2014) and several of them are regulated through miRNAs (i.e. miR 34-a). It would be interesting to study more extensively the relationship between p53 and GSNOR. On the contrary, data on mitophagy appear weak. As mentioned, the HEK293 cell model utilized to overexpress ATM mutants may not be perfectly suited here. In addition, CRISPR/Cas9 mediated ATM KO should be considered instead of shRNAs, immunofluorescence and western blots should be quantified and the authors should better characterize the purported decrease in mitochondria (i.e. flow cytometry or mitochondrial DNA).

e. Besides H₂O₂ induction, are other mitophagy inducers such as hypoxic stress inducing the same effects on GSNOR?

f. H₂O₂ also induces autophagy, not solely mitophagy. Is there a relationship between GSNOR

upregulation and autophagy induction?

g. What is the mechanisms through which GSNOR induces mitophagy? Does the induction rely on reported effectors such as FUNDC1, prohibitin2 or the PINK1-Parkin pathway?

h. Evidence of a connection between mitophagy and cell death is weak. Does the ability of GSNOR to rescue the inhibition of cell death depend on an increase in mitophagy?

i. It would be great to understand better the relationship between the ATM-GSNOR axis induced mitophagy and T-cell activation.

As of specific figures:

j. Fig 1: it seems that GSNOR mRNA and protein level would be reaching a peak after 8hr H₂O₂ treatment. Are GSNOR mRNA and protein level upregulated at earlier treatment times, perhaps 1 hr, 2 hr and 4hr? Is the phosphorylation of ATM and CHK2 occurring earlier than that GSNOR upregulation?

k. Fig 2C: why was CHK2 phosphorylation at Thr68 even higher after adding AZD since AZD is CHK2 inhibitor?

l. Fig 2E: GSNOR upregulation was not inhibited well. bad knockdown effect of CHK2 maybe the reason. Could you repeat the CHK2 Knockdown or knockout experiment to see GSNOR then?

m. How about the cellular ROS and mitochondrial ROS (Mito-ROS) level upon H₂O₂ 8hr treatment? Whether ROS/Mito-ROS could be an inducer for ATM-GSNOR upregulation?

n. Fig 3A could be moved to Fig 1 as it proves that H₂O₂ is inducing some damage at that specific concentration and time point.

o. Fig. 4 E, G. p53 may be wild-type is HeLa and U2OS, but it is functionally impaired by interaction with HPV-E6 and inactivation of p19 in those cells.

p. Fig 5A and Fig 6E: could you mark GSNOR level in "ATMCL + GSNOR" cells? It would be reasonable to calculate LC3-mitochondria co-localization in GSNOR positive ONLY cells if you want to do such rescue experiments.

a. Does GSNOR overexpression stimulate cell growth? In Fig 7C, there seem to be more cells in the GSNOR WT group, were the same number of cells seeded before treatment? Authors should make sure of that ahead of treating with compounds given the fact that sensitivity may depend on cell density.

b. Figure legends should be more detailed and informative.

Referee #2:

It is increasingly evident that oxidative stress-induced activation of ATM is highly relevant as a protective mechanism in the cell. The mechanism of activation of ATM by ROS is well described and distinct from that induced by DNA damage. At this stage signaling from ROS-activated ATM is much less well understood than that resulting from DNA double strand (DSB) breaks. This submission provides important new information on how ROS-activated ATM mediates the induction of the denitrosylase S-nitrosoglutathione reductase, GSNOR, at the translational level via pCHK2 and p-p53. In addition it describes its influence on mitophagy and cell survival. Finally it outlines a function for ROS/ATM/GSNOR in t-cell activation which is relevant to the immunodeficiency observed in patients lacking ATM function.

Specific comments

1. The submission provides new mechanistic data on activation of ATM-GSNOR by nitroxidative stress through induction of mitophagy. While a defect in mitophagy has already been described in ATM-deficient cells it provides further insight into another way on how this may be controlled (see comment below).

2. The mechanism is clearly outlined and supported by sound experimental results.
3. However, activation of ATM by low concentrations of H₂O₂ and the appearance of DNA damage (γH2AX) is confusing and does not appear to fit with data from the Paull lab (an author here) where 100μM H₂O₂ activates ATM without causing DNA damage. Needs to be explained. Stated later that DNA damage is not part of the mechanism??
4. ATM is activated rapidly by H₂O₂ so the relevance of GSNOR induction at 4-8h to biological function is unclear?
5. Fig 3 NCS induced phosphorylation of p53 but very high basal level of p53. I would have expected induction of p53 protein after NCS as with IR
6. The weaker phosphorylation of ATM in ATMCL compared to wt not that convincing but published previously.
7. Fig 4 use of Pifithrin-α inhibited p53 phosphorylation after H₂O₂ but p53 protein increased ??
8. It is somewhat difficult to understand how GSNOR induction protects cell viability by promoting mitophagy. This process is under complex control and one would expect several players as already published and not such a major role for GSNOR ??

Referee #3:

Review of 50500V1

In the manuscript by Cirotti et al., titled "Redox activation of ATM enhances GSNOR translation to sustain mitophagy and tolerance to oxidative stress", the authors investigate contribution of GSNOR in ATM-mediated defense of exogenous stress. Using pharmacologic and genetic tools, this study demonstrated clear connections between ATM, CHK2 and GSNOR in the prevention of oxidative stress. The authors also show the importance of p53 in this response. They extend these findings to implicate the ATM-GSNOR axis is guarding the cell against nitric oxide stress. Finally, they demonstrate the functional relevance of this protection in T cells. This is an excellent manuscript with well-supported conclusions. Only minor corrections are suggested.

Minor Comments

1. The overall flow of the figure panel layout has hard to follow. It is recommended that the figure panels (A, B, onwards) be arranged as increasing from left to right. There was an issue in Figure 1 and 5.
2. The authors state "Preincubation with KU55933 (KU) and AZD7762 (AZD), which were used to inhibit ATM and CHK2, respectively, significantly prevented GSNOR increase induced by H₂O₂ (Fig. 2B, C)", but in Fig. 2C, there does not appear to be a decrease in CHK2 phosphorylation at Thr68, even though GSNOR expression is reduced. Is this the correct blot that was intended to be used for this Figure panel?
3. The effect of ATM inhibition (KU) on T cell activation appears to be stronger than GSNOR inhibition (N6022) in Figure 8. Potential GSNOR-independent effects on T cell activation following ATM inhibition should be discussed.

Danish Cancer Society
Research Center

Strandboulevarden 49
DK-2100 Copenhagen
Denmark

Tel +45 3525 7500
Fax +45 3527 1811
www.cancer.dk

UNDER PROTECTION OF
HER MAJESTY THE QUEEN

July 25, 2020

Dear Editor,

Thank you for your letter regarding the manuscript EMBOR-2020-50500V1 entitled, “*Redox activation of ATM enhances GSNOR translation to sustain mitophagy and tolerance to oxidative stress*”.

We also want to thank the Reviewers for their careful reading of our paper: their comments and suggestions have been valuable to improve it. In this revised version, we made substantial changes in accordance to Reviewers’ recommendations and, for this reason, we have included one new co-author, who helped in performing new experiments, and added an equal contribution for Salvatore Rizza, who took care in designing, carrying out and rationalizing most of new analyses. A point-by-point reply to Reviewers’ comments (in bold) are as follows:

Reviewer #1:

...The authors point out that the ATM-GSNOR axis could be an oxidative stress sensor to fine-tune cell survival by linking together other quality control systems such as mitophagy. Their findings unveil an interesting role of ATM in the regulation of S-nitrosoglutathione reductase (GSNOR). Together with the well-known ability of GSNOR to regulate lymphocyte development, their finding could potentially contribute to explaining the lymphopenia observed in ataxia telangiectasia patients.

Although the mechanism proposed is interesting and quite novel, due to experimental weaknesses, several improvements are needed prior to eventually granting publication. Specifically,

a. I am concerned about the poor quality of the data, specifically of some of the western blots presented. Most if not all of them need quantification to be able to draw any solid conclusion.

We are honored that the Reviewer appreciated the novelty of our study and apologize with her/him for the poor quality of some Western blots present in the old version of the manuscript. As requested, we performed new analyses and/or quantitation of most of them in order give more strength to our results (please see below).

b. In Fig 2, the authors describe the dependency of GSNOR activation (defined as translational regulation in Fig 1) on ATM and CHK2. Yet of the inhibitors tested, KU effectively inhibits ATM phosphorylation, while AZD does not seem to inhibit efficiently CHK2 phosphorylation in Fig 2C, and yet, they both reduce GSNOR. Likewise, in Fig 2E (needs quantification by densitometry), CHK2 targeting siRNAs have very mild effect in silencing CHK2 and therefore in preventing its activation. Altogether, the regulatory effect of ATM on GSNOR seems interesting and plausible but the overall underlying mechanism is not completely clear and the proposed involvement of CHK2 is questionable.

We thank the Reviewer for her/his concerns regarding the phospho-levels of CHK2 shown in Fig. 2C and for giving us the opportunity to go into much detail in the mechanism of action of AZD7762 (AZD). Actually, AZD does not impede CHK2 from being phosphorylated at Thr68 by ATM, but binds the ATP binding pocket of the enzyme, thus acting as potent and selective ATP-competitive inhibitor. Based on this this, AZD should not modulate phospho-CHK2 levels (which, indeed, remain stable as shown in Fig. 2C), but inhibits phosphorylation catalyzed by CHK2 on its protein targets (e.g., p53, whose phosphorylation at Ser15 decreases significantly as shown in Fig. 4C). Basically, it is the same mode of action of the ATM inhibitor, KU55933 (KU). However, since ATM auto-phosphorylates, KU administration results (also) in a decrease of phospho-ATM immunoreactive band. In order to make this clear, we added one new sentence (see p.8, l.10 from the bottom)

Regarding the second issue – i.e., the very mild effects of CHK2 siRNA in downregulating CHK2 levels – we totally agree with the Reviewer and, in this new version of the manuscript, we show a new set of experiments with another siRNA which is more efficient. Please see new Western blot analyses in Fig. 2E, with densitometry of at least 3 Western blots from independent experiments.

c. The authors further show that NCS treatment does not induce GSNOR expression as well as oxidative stress (H₂O₂). Data shown in western blots should be quantified and accompanied by ROS quantification upon NCS treatment, by flow cytometry (CellRox or similar) to compare the amount of ROS produced by NCS or H₂O₂ treatment and exclude that the induction of DNA damage by NCS in this context generates as much oxidative stress as treatment with H₂O₂.

We thank the Reviewer for this very nice suggestion. In this new version of the manuscript, we have added Western blot densitometries, and new FACS analyses of ROS production upon incubation with NCS (new Fig. 3E) and H₂O₂ (new Fig. 5A). The results obtained indicate that NCS treatment is not associated with H₂O₂ production (measured upon incubation with 2'7'-H₂DCF-DA), this reinforcing the hypothesis that GSNOR induction is responsive to oxidative stress and not to DNA damage.

...Similarly, data in Fig 3E are misleading, as in the "empty" lane H₂O₂ does not seem as effective as in the other experiments in inducing ATM phosphorylation; nevertheless, GSNOR appears to be effectively induced. In addition, it is not clear why the overexpression of wild type ATM would increase GSNOR expression, making it very hard to evaluate its phosphorylation in the western blots presented (Fig 3E) without proper quantification

We apologize with the Reviewer if Fig. 3E (new Fig. 3F) led to a misinterpretation of our results. However, it is important to point out that her/his criticism derives mostly from the impossibility to show, in a unique Western blot, how ATM phosphorylation is differently modulated in cells expressing different levels of ATM. A single exposure time (like that selected in the previous version of the manuscript) was aimed at making the reader appreciate differences of phospho-ATM levels in cells overexpressing *ATM*^{WT} and *ATM*^{CL}, but, unfortunately, flattened those produced, at the endogenous level, by H₂O₂ in *Empty* cells. Had we increased the exposition time to make this clearer, we would have saturated phospho-ATM signal in ATM overexpressing cells. To meet Reviewer's request, in the new version of the manuscript we have performed densitometric analyses, and shown two exposure times (short and long) of the same Western blot in order to better emphasize phospho-ATM modulation by H₂O₂ in all the cell systems used.

Regarding the basal increase of GSNOR levels observed upon ATM overexpression, we think this is in line with our hypothesis that ATM activity triggers the induction of GSNOR. It worth to mention that, a basal increased activation of ATM is frequently detected upon ATM overexpression. This is coherent with the fact that, in these conditions, molecular repression mechanisms aimed at keeping ATM inactivated without stimuli are partly circumvented. Similarly, in Fig. 3F, it is evident that reconstitution of ATM with *ATM*^{WT} results in ATM activation which in turn leads to an increase of GSNOR levels, as pointed out by the Reviewer. Consistently with all other results, H₂O₂ further increases ATM activity, thus enhancing GSNOR expression.

d. The involvement of ATM and p53 in the regulation of GSNOR presented in Fig. 4 and the correlation with the survival advantage provided by this axis in T-cells (Fig 8) is perhaps the most intriguing part of the manuscript but remains poorly explored. Indeed, at the translational level P53 mostly regulates negatively other effectors (i.e. those involved in the cell cycle). The authors here propose that ATM/P53 axis positively regulates at translational level GSNOR. It is likely that GSNOR regulation by p53 is

indirect. Several studies have already identified other p53 translational targets (e.g. Zaccara et al, Cell Death Differ., 2014) and several of them are regulated through miRNAs (i.e. miR 34-a). It would be interesting to study more extensively the relationship between p53 and GSNOR.

We are really glad that the Reviewer has appreciated this part of our study, and fully agree with her/him that the relationship between p53 and GSNOR would deserve to be more extensively investigated. In this regard, we are seeking to be as comprehensive as possible and planning to study different possible mechanisms underlying this interplay (including the link between S-nitrosylation status and p53 glycosylation). However, data are still few and preliminary and would need an *ad hoc* study to be fully defined. The Reviewer will agree with us that, although interesting, this aspect is marginal to this paper and beyond the main scope of our study.

...On the contrary, data on mitophagy appear weak. As mentioned, the HEK293 cell model utilized to overexpress ATM mutants may not be perfectly suited here. In addition, CRISPR/Cas9 mediated ATM KO should be considered instead of shRNAs

We thank the Reviewer for this comment and for giving us the opportunity to better elucidate some technical details about how these cells are generated and why they are a good model for our experiments.

U2OS cells used in our study are those generated, set up, and already published by our team (i.e., by Tanya T. Paull's Research Group). Please see Zhang et al. (*Sci. Signal.* 2018 11(538). pii: eaaq0702) for detailed description. Basically, these cells are U2OS Flp-In T-Rex, which carry the alleles of WT, 2RA, or CL forms of ATM under the control of a doxycycline responsive promoter. Before inducing their expression by doxycycline addition, cells must be depleted of endogenous ATM by means of repeated transfections with short hairpin (sh) RNAs against ATM (every 48 h for three consecutive times). This is required to minimize:

- i) any effects deriving from keeping the endogenous ATM in cells expressing the mutant forms of the protein (2RA or CL), which would hide the effects of the mutants;
- ii) differences in expression rate, as – being responsive to doxycycline induction – all the ectopic forms of ATM (WT, 2RA or CL) are uniformly transcribed.

Based on the above, ATM^{WT} cells are NOT simply parental U2OS cells, but they are U2OS Flp-In T-Rex in which we depleted the endogenous ATM (WT by definition) and replenished them with another WT form of the protein. In principle they are WT but, thus generated, they are comparable with cells that express the redox insensitive mutant ATM^{CL} or the DNA-damage “unresponsive” ATM^{2RA}.

That said, the answer to the Reviewer's question about “why we have preferred these systems and did not generate ATM-KO cells by CRISPR/Cas9 technology”, deals essentially with the fact that:

- i) During clonal selection, CRISPR/Cas9-generated cells could develop adaption to ATM ablation. Reasonably, this would result in enhanced resistance to DNA damage and increased mutation rate, both affecting the reliability of our results.

ii) This would impose to generate ATM mutants (2RA and CL) by the same technology, for uniformity. As above argued for ATM-KO cells, also in this case, there would be the possibility that clonal selection of cells carrying these point mutations may compromise the reliability of our results. This takes on even more importance if we consider that ATM^{2RA} mutant is not responsive to DNA damage and its expression would make the cells more inclined to accumulate mutations.

...immunofluorescence and western blots should be quantified and the authors should better characterize the purported decrease in mitochondria (i.e. flow cytometry or mitochondrial DNA).

We thank the Reviewer for this suggestion and agree with her/him that some aspects of mitophagy were not properly investigated, especially in HEK cells. For this reason, we performed new sets of experiments in which HEK cells were treated with H₂O₂ or CCCP and mitophagy was evaluated by:

- 1) Western blot analyses of at least 3 mitochondrial proteins spanning both inner and outer mitochondrial membranes;
- 2) qRT-PCR of D-loop, which is a measure of mtDNA and, indirectly, an esteem of mitochondrial mass.

Confocal microscopy analyses were not included due to reduced cytoplasm of HEK cells which does not allow to evaluate mitophagy. This technique was used only in U2OS cells that, in the new version of the manuscript, have also been analysed for mtDNA content (for uniformity with HEK cells). Indeed, as requested, we have included quantitations of Western blot and performed qRT-PCR analyses of mtDNA, both upon H₂O₂- and CCCP-treatment (new Fig. 5F and 6D).

As for immunofluorescence quantitation, we want to remind to the Reviewer that we had already calculated the rate of mitophagy (new Figs. 5G and 6F) as % of mitochondrial particles (red fluorescence) co-localizing with LC3 puncta/cell (green). According to what reported in Materials and Methods section, we made use of Fiji analysis software using the open-source plugin ComDet v. 0.3.7 that analyzes colocalization of bright intensity spots in images with heterogeneous background.

e. Besides H₂O₂ induction, are other mitophagy inducers such as hypoxic stress inducing the same effects on GSNOR?

We thank the Reviewer for this very nice suggestion. As requested, we have performed new experiments to give strength to the hypothesis that ATM/GSNOR axis activation is a ROS dependent pro-mitophagic signaling pathway. To this end, we treated HEK cells with a combination of antimycin and oligomycin (AO), or subjected them to hypoxia (i.e., 1% pO₂). Both these conditions are used to induce mitophagy but are not associated with ROS generation. Consistently, we observe a decrease of mitochondrial proteins and mtDNA which is not accompanied by ATM/GSNOR axis activation and ROS production. These results are now shown in new Fig. EV2

f. H₂O₂ also induces autophagy, not solely mitophagy. Is there a relationship between GSNOR upregulation and autophagy induction?

We thank the Reviewer for the opportunity to discuss better this aspect. She/He is perfectly right when she/he says that H₂O₂ induces autophagy in general. Actually, it has been quite recently demonstrated that H₂O₂ – at the same concentrations we also used in our paper – induces pexophagy directly *via* ATM through phosphorylation of PEX5. Mitochondria, like peroxisomes, are other intracellular sources of ROS, therefore, from this perspective, ATM could be considered as a very upstream sensor of ROS that orchestrates the selective removal of ROS-producing organelles (i.e., mitochondria and peroxisomes) as general protective response to oxidative stress.

Notwithstanding this new pro-autophagic role of ATM, up to date, we have no data arguing for a direct relationship between GSNOR and autophagy. Several lines of evidence that we accumulated in the last five years indicate that GSNOR-null animals and cells do not show any specific deficit of bulk autophagy, suggesting that GSNOR does not affect autophagy *per se*. However, we demonstrated that GSNOR-deficiency results in defective mitochondrial dynamics and mitophagy due to the hyper *S*-nitrosylation of Drp1 and Parkin. This translates in accumulation of highly fragmented mitochondria (in response to the hyperactivation of Drp1) which are not properly recognized by the autophagic machinery (caused by the inhibition of Parkin ubiquitylating activity). Therefore, the ATM-mediated induction of GSNOR, which we detected upon H₂O₂ treatment, is aimed at pushing mitophagy by denitrosylating Parkin, as well as other molecular factors involved in this process, e.g. PINK1, which are known to be inhibited by *S*-nitrosylation (Oh et al., *Cell Rep.* 2017). Fig. 2F confirms this hypothesis, as the basal level of SNO-proteins significantly decreases after H₂O₂ treatment, standing for denitrosylation being efficiently induced. Moreover, NO and H₂O₂ increase, which is detected upon T cell stimulation – reasonably as a response to the well documented activation of NOS and NOX, respectively – provides an additional cue that cellular denitrosylating capacity is indispensable to regulate mitophagy rate in response to nitrosative and oxidative insults.

g. What is the mechanisms through which GSNOR induces mitophagy? Does the induction rely on reported effectors such as FUNDC1, prohibitin2 or the PINK1-Parkin pathway?

Recalling what mentioned in **point f**, our recent literature indicates that mitophagy is sustained by GSNOR through the denitrosylation of key proteins, namely Parkin (Rizza et al., 2018, *PNAS*; Rizza et al., 2020, *Biochem. Pharmacol.*), which is well-documented to be inhibited by *S*-nitrosylation. Obviously, we cannot exclude *a priori* that other mitophagy-related proteins are also affected by GSNOR denitrosylating activity, provided that they are sensitive to *S*-nitrosylation. So far, there is no evidence that FUNDC1 is target of *S*-nitrosylation. Conversely, it has very recently been published that prohibitin 2 (PHB2) is *S*-nitrosylated at Cys69 (Qu et al., 2020, *J. Neurosci.*), this resulting in neuro-protection.

Although there is no evidence regarding the mechanism(s) through which this effect is induced, we cannot rule out they can depend on mitophagy.

However, on the basis of our recent literature, and given the new results obtained upon Parkin overexpression (see new Fig. 7D and next **point h**), we still believe – and hope the Reviewer agrees on this – that Parkin denitrosylation has a major role in GSNOR-induced mitophagy.

h. Evidence of a connection between mitophagy and cell death is weak. Does the ability of GSNOR to rescue the inhibition of cell death depend on an increase in mitophagy?

We thank the Reviewer for this comment. In line with what explained in **point g** about the major role of Parkin in regulating GSNOR-induced mitophagy, new results shown in Fig. 7D confirm that GSNOR overexpression exerts protection to H₂O₂ and NO-induced cell death only in the presence of Parkin. Indeed, siRNA against Parkin abolishes the protective effects of GSNOR, indicating that Parkin-mediated mitophagy is the main process responsible for this phenomenon.

i. It would be great to understand better the relationship between the ATM-GSNOR axis induced mitophagy and T-cell activation.

We understand the wish of the Reviewer, and we also agree on the fact that it would be fascinating to dissect the relationship between ATM and GSNOR, and if this axis plays a role in T-cell activation *via* the induction of mitophagy. Unfortunately, the COVID-19 pandemics has made some analyses more difficult to be performed; mostly, those on human samples, no matter if allowed by the Ethic committee. Access to the hospital and use of blood samples was not permitted, therefore it has been basically impossible for us to get new data to address Reviewer's curiosity. We hope that the Reviewer will understand.

As of specific figures:

j. Fig 1: it seems that GSNOR mRNA and protein level would be reaching a peak after 8hr H₂O₂ treatment. Are GSNOR mRNA and protein level upregulated at earlier treatment times, perhaps 1 hr, 2 hr and 4hr?

Is the phosphorylation of ATM and CHK2 occurring earlier than that GSNOR upregulation?

We thank the Reviewer for this comment and, as requested, we performed new experiments at very early time points (1-to-4 h). In Fig. 1 D, we now add GSNOR mRNA level analyses from 2 to 24 h, which show no changes. Results from Western blot analyses presented in new Fig. EV1 show that ATM/CHK2 axis (as well as ATR/CHK1) is rapidly induced upon H₂O₂, whereas GSNOR levels begin to increase at 4 h. These new set of data together with those provided in Fig. 3, support the hypothesis that GSNOR induction downstream of the persistent activation of ATM/CHK2 signaling axis (i.e., up to 24 h after H₂O₂ administration) is not related to DNA damage.

k. Fig 2C: why was CHK2 phosphorylation at Thr68 even higher after adding AZD since AZD is CHK2 inhibitor?

As mentioned in **point b**, AZD does not inhibit CHK2 from being phosphorylated at Thr68 by ATM, but binds the ATP-binding pocket, this stopping the phospho-signal from being transduced. Based on this, the main effect of AZD is to prevent phosphorylation of CHK2 protein targets, but, in these conditions, ATM-mediated phosphorylation – which is upstream CHK2 – is still “ON” as it does not receive any negative feedback from downstream effectors. It can happen that, when a signal is blocked and not able to be transduced in a coherent cell response, players upstream the blockage result to be equal (or even more) phosphorylated, this, however, not standing for an increased output of the signal. In order to make this clear, we added one new sentence (see p.8, l.10 from the bottom). Anyway, we selected another Western blot that is now more coherent with the densitometry calculated on 3 different experiments.

l. Fig 2E: GSNOR upregulation was not inhibited well. bad knockdown effect of CHK2 maybe the reason. Could you repeat the CHK2 Knockdown or knockout experiment to see GSNOR then?

We apologize with the Reviewer for the poor quality of this set of results, and agree with her/him that this part should be improved. As requested we used another (more effective) CHK2 siRNA and performed again the experiments. New results are now shown in Fig. 2E.

m. How about the cellular ROS and mitochondrial ROS (Mito-ROS) level upon H₂O₂ 8hr treatment? Whether ROS/Mito-ROS could be an inducer for ATM-GSNOR upregulation?

We thank the Reviewer for this acute suggestion. We agree with her/him that these data were missing in the previous version of the manuscript. Therefore, as requested, we evaluated H₂O₂ (with 2'7'-H₂DCF-DA) and mitochondrial superoxide (with Mito-SOX) upon H₂O₂ treatment over time (i.e., at 2, 4 and 8 h). Results now shown in Figs. 5A and 5B indicate that both ROS are generated soon after 2 h, but accumulate time-dependently, suggesting they are produced inside mitochondria, with H₂O₂, required for ATM/GSNOR axis activation, reasonably generated upon superoxide dismutation.

n. Fig 3A could be moved to Fig 1 as it proves that H₂O₂ is inducing some damage at that specific concentration and time point.

We understand the Reviewer's suggestion. However, as this new version of the manuscript is conceived, we prefer to gather all the results obtained on DNA damage in a single figure (Fig. 3) and, along with those shown in Fig. EV1, be more convincing in supporting the hypothesis that ATM/CHK2/GSNOR axis is activated by a redox mechanism.

o. Fig. 4 E, G. p53 may be wild-type is HeLa and U2OS, but it is functionally impaired by interaction with HPV-E6 and inactivation of p19 in those cells.

We thank the Reviewer her/his comment, which certainly deserves some explanations. As she/he pointed out, indeed, HeLa cells express a wild-type form of p53, most of which is maintained inactive by HPV16 E6 tumor antigen (HeLa). Conversely, U2OS has been previously reported to be wild-type for both p53 and Rb (Diller et al., 1990, *Mol. Cell. Biol.* 10: 5772–5781; Isfort et al., 1995, *Mol. Carcinog.* 14: 170–178; Allan and Fried, 1999, *Oncogene* 18: 5403–5412). Anyway, although HeLa cells are partially compromised for p53 activity, they do still contain a fully functional p53 pathway (Scheffner et al., 1990, *Cell* 63: 1129; Karlseder et al., 1999, *Science* 283: 1321–5). Therefore, despite the fact that the p53 pathway might be disrupted at some level, these cancer-derived cell lines are still widely used to study p53 activities, since it is still partly functional. Consistently, GSNOR induction in HeLa cells is not as pronounced as, instead, is in the other cell lines used.

p. Fig 5A and Fig 6E: could you mark GSNOR level in "ATMCL + GSNOR" cells? It would be reasonable to calculate LC3-mitochondria co-localization in GSNOR positive ONLY cells if you want to do such rescue experiments.

We understand Reviewer's concern and thank her/him for giving us the opportunity to better explain the methodological aspects of these experiments. Cells shown Figs. 5A (now Fig. 5H) and 6E represent those incorporating the plasmid coding for (and efficiently expressing) ATM shRNA. This plasmid carries also GFP as a reporter gene, whose expression is required to provide the efficiency of transfection (in our conditions being quite high, around 90%; please look at GSNOR and P-ATM immunoreactive bands in new Fig. 6C) and recognize only those cells that incorporated shATM-carrying plasmid by fluorescence microscopy. In fact, cells displayed in previous Fig. 5A and 6E are fluorescent and were marked as "shATM".

In some experiments, shATM-coding plasmid is co-transfected with the one carrying GSNOR. In cases of co-transfections, it is commonly accepted that the transfection complex (in our case generated by Genejuice from Merck®) always contains a representative mixture of both plasmids, which will be equally delivered into the same cells. This assumption implies that, if we detect GFP expression (like in the cells shown in new Fig. 5H and 6E), we can be reasonably confident that these cells equally incorporated shATM and GSNOR-coding plasmids. Based on this Reviewer's concern, in the new version of the manuscript, we have substituted "shATM" cells with "transfected cells", hoping this helps understand that these cells express both shATM and GSNOR.

To further convince the Reviewer that GSNOR ectopic expression is able to rescue mitophagy in ATM^{CL} cells, we show Western blot of mitochondrial proteins and (in the new version of the manuscript) qRT-PCR of mtDNA. Although both these analyses are performed in a mixed population composed by transfected and non-transfected cells – according to our estimations in proportion of 90:10 – the significant decrease observed in the levels of mitochondrial proteins and mtDNA gives further support to the reliability of our experimental model and the efficiency of co-transfection.

a. Does GSNOR overexpression stimulate cell growth? In Fig 7C, there seem to be more cells in the GSNOR WT group, were the same number of cells seeded before treatment? Authors should make sure of that ahead of treating with compounds given the fact that sensitivity may depend on cell density.

We thank the Reviewer for this nice catch. Actually, we have never experienced any change in the rate of proliferation induced upon GSNOR overexpression, and the number of cells seeded is always the same in each experiment (i.e., $4 \times 10^4/\text{cm}^2$). However, to convince the Reviewer of our experimental setup, we measured, by Alamar blue (AB) staining, the number of viable cells in the 24 hours after transfection – which is the time selected for this set of experiments – and graphed the values of fluorescence as it shown below. As also the Reviewer can appreciate, no significant change is observed.

b. Figure legends should be more detailed and informative.

We apologize with the Reviewer for missing information. In the new version of the manuscript we have modified the figure legends as requested.

Reviewer #2:

...This submission provides important new information on how ROS-activated ATM mediates the induction of the denitrosylase S-nitrosoglutathione reductase, GSNOR, at the translational level via pCHK2 and p-p53. In addition, it describes its influence on mitophagy and cell survival. Finally, it outlines a function for ROS/ATM/GSNOR in t-cell activation which is relevant to the immunodeficiency observed in patients lacking ATM function.

We are glad and honored that Reviewer 2 appreciated our work. This is a great source of pride for us.

Specific comments

1. The submission provides new mechanistic data on activation of ATM-GSNOR by nitroxidative stress through induction of mitophagy. While a defect in mitophagy has already been described in ATM-deficient cells, it provides further insight into another way on how this may be controlled (see comment below).

2. The mechanism is clearly outlined and supported by sound experimental results.

3. However, activation of ATM by low concentrations of H₂O₂ and the appearance of DNA damage (γ H2AX) is confusing and does not appear to fit with data from the Paull lab (an author here) where 100 μ M H₂O₂ activates ATM without causing DNA damage. Needs to be explained. Stated later that DNA damage is not part of the mechanism??

We thank the Reviewer for this comment and agree with her/him that we probably did not stress enough the message in the previous version of our manuscript. Actually, our results – in agreement with what demonstrated by Paull's lab in the past – show that H₂O₂ (at the low concentration used in our study) does not induce any severe, but just a mild DNA damage which is rapidly repaired.

It is also worth to note that, from a mere chemical point of view, H₂O₂ is a mild oxidant. This means that it mostly produces DNA single strand breaks: a kind of damage repaired *via* ATR in a process that does not involve ATM. Therefore, we can conclude that the persistent activation of ATM (up to 24 h) detected in our conditions is not the result of DNA damage, but consequent to redox modification initiated by H₂O₂ and maintained by mitochondrial ROS.

As requested by the Reviewer, we have provided new results (see new Fig. 3 and EV1), supporting the hypothesis that the mild DNA damage detected upon H₂O₂ treatment is not part of the mechanism.

4. ATM is activated rapidly by H₂O₂ so the relevance of GSNOR induction at 4-8h to biological function is unclear?

We thank the Reviewer for giving us the opportunity to clarify this aspect. As above discussed (point 3), ATM is maintained in the active state from 1 to 24 h after H₂O₂ treatment to sustain an increased mitophagy rate *via* GSNOR. However, our results argue for GSNOR being not stabilized by ATM, but, actually, induced via a translational regulation through p53 in a still not defined way. This suggests that the effects of ATM activation on GSNOR levels are not fast (in the range of minutes) but slower (hours) as expected from a modulation in the rate of translation. As shown in new Fig. EV1B, GSNOR increase begins to be significant not earlier than 4 h of treatment.

5. Fig 3 NCS induced phosphorylation of p53 but very high basal level of p53. I would have expected induction of p53 protein after NCS as with IR

We thank the Reviewer for this nice catch and apologize for inaccuracy. As suggested, we selected a new Western blot in which p53 expression levels are increased as expected, and showed densitometry of all the Western blots performed (Please, see new panel in Fig. 3D).

6. The weaker phosphorylation of ATM in ATM^{CL} compared to wt not that convincing but published previously.

We understand Reviewer's concern, and confirm that, as previously reported, ATM phosphorylation levels detected in *ATM^{CL}* cells is weaker than those observed in *ATM^{WT}* cells. Actually, a basal increased activation of ATM is frequently detected upon ATM overexpression. This is coherent with the fact that, in these conditions, molecular repression mechanisms aimed at keeping ATM inactivated without stimuli are partly circumvented. Based on this, we might speculate that Cys2991 could not be only required for ROS-mediated activation of ATM, but also play a role in the processes underlying the generation and/or stabilization of the phospho-active isoform of the enzyme. According to this hypothesis, the *ATM^{CL}* mutant should be in general less efficient in mediating/maintaining ATM autophosphorylation.

7. Fig 4 use of Pifithrin- α inhibited p53 phosphorylation after H₂O₂ but p53 protein increased ??

We understand the Reviewer's concern. However, p53 phosphorylation can be frequently accompanied by an increase of its expression levels, a phenomenon that is amplified and more visible when the signal is inhibited and cannot be transduced (as, in this case, by pifithrin- α).

8. It is somewhat difficult to understand how GSNOR induction protects cell viability by promoting mitophagy. This process is under complex control and one would expect several players as already published and not such a major role for GSNOR??

We thank the Reviewer for giving us the possibility to better explain this issue. We agree with her/him that many players are involved in this process. However, in agreement with our previous papers (Rizza et al., *PNAS* 2018; Rizza et al., *Biochem. Pharmacol.* 2020) and new results obtained upon Parkin silencing (see new Fig. 7D), we believe that GSNOR, *via* Parkin denitrosylation, increases mitophagy rate and, in turn, primarily contributes to cell survival towards nitro-oxidative stress.

Reviewer #3:

... This is an excellent manuscript with well-supported conclusions. Only minor corrections are suggested.

We thank the Reviewer for her/his nice words and very positive evaluation of our manuscript.

Minor Comments

1. The overall flow of the figure panel layout has hard to follow. It is recommended that the figure panels (A, B, onwards) be arranged as increasing from left to right. The was an issue in Figure 1 and 5.

We apologize with the Reviewer for the misleading order of the panels. We have now tried to do all our best to fit everything in place, maintaining the correct flow of panel layout.

2. The authors state "Preincubation with KU55933 (KU) and AZD7762 (AZD), which were used to inhibit ATM and CHK2, respectively, significantly prevented GSNOR increase induced by H₂O₂ (Fig. 2B, C)", but in Fig. 2C, there does not appear to be a decrease in CHK2 phosphorylation at Thr68, even though GSNOR expression is reduced. Is this the correct blot that was intended to be used for this Figure panel?

We thank the Reviewer for her/his concerns regarding the phospho-levels of CHK2 shown in Fig. 2C. As explained to Reviewer 1, AZD does not prevent CHK2 phosphorylation at Thr68 by ATM, but binds the ATP binding pocket of the enzyme, thus acting as potent and selective ATP-competitive inhibitor. Based on this this, AZD should not modulate phospho-CHK2 levels (which, indeed, remain stable as shown in Fig. 2C), but inhibits phosphorylation of CHK2 protein targets (e.g., p53, whose phosphorylation at Ser15 decreases significantly as shown in Fig. 4C). In order to make this aspect clear, we added one new sentence (see p.8, l.10 from the bottom)

3. The effect of ATM inhibition (KU) on T cell activation appears to be stronger than GSNOR inhibition (N6022) in Figure 8. Potential GSNOR-independent effects on T cell activation following ATM inhibition should be discussed.

We thank the Reviewer for this observation, which is even more relevant if one looks at the effects produced by ATM inhibition on T cell activation (results shown in Fig. 8H). This phenomenon is probably due to the fact that ATM plays multiple roles in cell homeostasis, the most important of which is to guard any possible defects/damage in DNA, such as those occurring when cell actively replicate. By definition, blast activation implies T cell enter rapid replication phase. Therefore, it is reasonable that ATM inhibition does not only affect GSNOR-mediated potentiation of mitophagy, but rather impact broadly the capacity/efficiency of replication. As requested by the Reviewer, we have elaborated on this in the last sentences of the Results.

We hope that You and the Reviewers can appreciate the efforts we made to improve the manuscript and be now convinced that our results are relevant and robust enough to be published in *EMBO Reports*,

Yours sincerely,

Giuseppe Filomeni, PhD

Dear Dr. Filomeni

Thank you for the submission of your revised manuscript to EMBO reports. Your manuscript was sent back to former referee 1 and 2 and we have now received their reports (copied below).

As you can see, the referees find that you have addressed all concerns during revision and recommend publication.

Browsing through the manuscript myself, I noticed a few editorial things that we need before we can proceed with the official acceptance of your study.

1) Funding information: Please add PRIN-2015LZE9944 to the relevant section in our online submission system.

2) Please add a Data availability section at the end of Materials & Methods. You can state that you have no data that requires deposition in a public database.

3) Appendix

- Please add a title page with a table of content (incl. page numbers)
- Appendix Figure S3: Please describe the number of cells analysed (n) in the figure legend.
- Appendix Figure S4: please add information on the statistical test used and please state whether n = 3 refers to biological or technical replicates

4) Figures/Source data

- Figure 3B and 7H: during our routine image integrity analysis that we perform on all revised manuscripts we noticed that the Western blot data shown in Figure 3B and 7H displays long vertical lines. I attach a screen shot of one of these so that you can see what we mean. Since these were not present in the source data you supplied they are likely artefacts that occurred during figure preparation and/or compression. I suggest redoing these two panels to get rid of these artefacts. I also noticed that the source data for Fig. 7H is missing. Could you please supply it?

- The source data you supplied for the GSNOR blot in Figure 4C appears not to match the data shown in the panel. Can you please doublecheck these data and supply the correct source data for this blot?

- Source data for P-CHEK1(Ser317) and CHEK1, Figure EV1A, upper panel: it appears that you highlight the incorrect lanes in the source data. Looking at the bands and the background you appear to display lanes 3 - 8 in the figure panel instead of the boxed lanes 1 -6. Please carefully check the data shown in the figure panel and the source data and verify that the correct samples have been chosen and that the samples/lanes shown match the experimental conditions indicated in the figure (H₂O₂ treatment or not).

- Figure 3C, GAPDH panel: the GAPDH blot in the panel does not quite match the blot shown in the source data, where the much lower levels of GAPDH in column 4 (HU) are more obvious. I suggest using as little contrast modification as possible to represent the data in the most accurate manner.

- Please split the source data into one file per figure.

5) Figures:

- Figure 3A, 5H, 6E, EV4A: please change the unit of the scale bar from mm to micrometer in these panels.

- The following panels need scale bars:

** Figure 3a - Please add zoom box scale bars

** Figure 5h - Please add merge scale bars

** Figure 6E - Please add merge zoom boxes scale bars

** Figure EV4 A - Please add merge zoom boxes scale bars

** Figure 7H needs scale bars

6) Figure legends:

- There are some remaining issues you need to correct in the figure legends:

- Figure 3B: please define the nature of the error bars, e.g. SD

- Figure 5G, H, Figure 6F, Figure EV4B: please define the central band, boxes and whiskers of the boxplot. The description "means plus/minus SD" does not apply to boxplots.

- Figure 7H: please add scale bars to this panel and define their size in the figure legend.

- Figure 8A, B, D, G: please define the box limits

- Figure EV3E, G: please define the bars and error bars, e.g. mean plus/minus SD and the number of replicates

With kind regards,

Referee #1:

The manuscript is in my opinion suitable for publication.

Referee #2:

The authors have addressed all the issues that I raised and carried out additional experimentation. I am pleased to recommend publication in EMBO Reports.

The authors have addressed all minor editorial requests.

Dr. Giuseppe Filomeni
Danish Cancer Society Research Center
Redox Signaling and Oxidative Stress
Strandboulevarden 49
Copenhagen 2100
Denmark

Dear Giuseppe,

I am very pleased to accept your manuscript for publication in the next available issue of EMBO reports. Thank you for your contribution to our journal.

At the end of this email I include important information about how to proceed. Please ensure that you take the time to read the information and complete and return the necessary forms to allow us to publish your manuscript as quickly as possible.

As part of the EMBO publication's Transparent Editorial Process, EMBO reports publishes online a Review Process File to accompany accepted manuscripts. As you are aware, this File will be published in conjunction with your paper and will include the referee reports, your point-by-point response and all pertinent correspondence relating to the manuscript.

If you do NOT want this File to be published, please inform the editorial office within 2 days, if you have not done so already, otherwise the File will be published by default [contact: emboreports@embo.org]. If you do opt out, the Review Process File link will point to the following statement: "No Review Process File is available with this article, as the authors have chosen not to make the review process public in this case."

Should you be planning a Press Release on your article, please get in contact with emboreports@wiley.com as early as possible, in order to coordinate publication and release dates.

Thank you again for your contribution to EMBO reports and congratulations on a successful publication. Please consider us again in the future for your most exciting work.

Kind regards,
Martina

Martina Rembold, PhD
Editor
EMBO reports

THINGS TO DO NOW:

You will receive proofs by e-mail approximately 2-3 weeks after all relevant files have been sent to

our Production Office; you should return your corrections within 2 days of receiving the proofs.

Please inform us if there is likely to be any difficulty in reaching you at the above address at that time. Failure to meet our deadlines may result in a delay of publication, or publication without your corrections.

All further communications concerning your paper should quote reference number EMBOR-2020-50500V3 and be addressed to emboreports@wiley.com.

Should you be planning a Press Release on your article, please get in contact with emboreports@wiley.com as early as possible, in order to coordinate publication and release dates.

Corresponding Author Name: Giuseppe Filomeni

Manuscript Number: EMBOR-2020-50500V1